# Chimeric origins of ochrophytes and haptophytes revealed through an ancient plastid proteome

Richard G Dorrell[1]*, Gillian Gile[2], Giselle McCallum[1], Raphaël Méheust[3], Eric P Bapteste[3], Christen M Klinger[4], Loraine Brillet-Guéguen[5], Katalina D Freeman[2], Daniel J Richter[6,7], Chris Bowler[1]*

[1]IBENS, Département de Biologie, École Normale Supérieure, CNRS, Inserm, PSL Research University, Paris, France; [2]School of Life Sciences, Arizona State University, Tempe, United States; [3]Institut de Biologie Paris-Seine, Université Pierre et Marie Curie, Paris, France; [4]Department of Cell Biology, University of Alberta, Edmonton, Canada; [5]CNRS, UPMC, FR2424, ABiMS, Station Biologique, Roscoff, France; [6]Sorbonne Universités, Université Pierre et Marie Curie, CNRS UMR 7144; [7]Adaptation et Diversité en Milieu Marin, Équipe EPEP, Station Biologique de Roscoff, Roscoff, France

*For correspondence: dorrell@
biologie.ens.fr (RGD); cbowler@
biologie.ens.fr (CB)

**Competing interests:** The authors declare that no competing interests exist.

**Abstract** Plastids are supported by a wide range of proteins encoded within the nucleus and imported from the cytoplasm. These plastid-targeted proteins may originate from the endosymbiont, the host, or other sources entirely. Here, we identify and characterise 770 plastid-targeted proteins that are conserved across the ochrophytes, a major group of algae including diatoms, pelagophytes and kelps, that possess plastids derived from red algae. We show that the ancestral ochrophyte plastid proteome was an evolutionary chimera, with 25% of its phylogenetically tractable nucleus-encoded proteins deriving from green algae. We additionally show that functional mixing of host and plastid proteomes, such as through dual-targeting, is an ancestral feature of plastid evolution. Finally, we detect a clear phylogenetic signal from one ochrophyte subgroup, the lineage containing pelagophytes and dictyochophytes, in plastid-targeted proteins from another major algal lineage, the haptophytes. This may represent a possible serial endosymbiosis event deep in eukaryotic evolutionary history.

## Introduction

Since their origin, the eukaryotes have diversified into an extraordinary array of organisms, with different genome contents, physiological properties, and ecological adaptations (*Dorrell and Smith, 2011*; *de Vargas et al., 2015*; *Dorrell and Howe, 2012a*). Perhaps the most profound change that has occurred within individual eukaryotic cells is the acquisition of plastids via endosymbiosis, which has happened at least eleven times across the tree of life (*Dorrell and Smith, 2011*). All but one characterized group of photosynthetic eukaryotes possess plastids resulting from a single ancient endosymbiosis of a beta-cyanobacterium by an ancestor of the archaeplastid lineage (consisting of green algae and plants, red algae, and glaucophytes) (*Dorrell and Smith, 2011*).

Photosynthesis has subsequently spread outside of the archaeplastids through secondary, tertiary, or more complex endosymbiosis events. By far the most ecologically successful of these lineages are those that possess plastids derived from secondary or more complex endosymbioses of a red alga (*Dorrell and Smith, 2011*; *Baurain et al., 2010*; *Stiller et al., 2014*). These are the 'CASH

**eLife digest** The cells of most plants and algae contain compartments called chloroplasts that enable them to capture energy from sunlight in a process known as photosynthesis. Chloroplasts are the remnants of photosynthetic bacteria that used to live freely in the environment until they were consumed by a larger cell. "Complex" chloroplasts can form if a cell that already has a chloroplast is swallowed by another cell.

The most abundant algae in the oceans are known as diatoms. These algae belong to a group called the stramenopiles, which also includes giant seaweeds such as kelp. The stramenopiles have a complex chloroplast that they acquired from a red alga (a relative of the seaweed used in sushi). However, some of the proteins in their chloroplasts are from other sources, such as the green algal relatives of plants, and it was not clear how these chloroplast proteins have contributed to the evolution of this group.

Many of the proteins that chloroplasts need to work properly are produced by the host cell and are then transported into the chloroplasts. Dorrell et al. studied the genetic material of many stramenopile species and identified 770 chloroplast-targeted proteins that are predicted to underpin the origins of this group. Experiments in a diatom called *Phaeodactylum* confirmed these predictions and show that many of these chloroplast-targeted proteins have been recruited from green algae, bacteria, and other compartments within the host cell to support the chloroplast.

Further experiments suggest that another major group of algae called the haptophytes once had a stramenopile chloroplast. The current haptophyte chloroplast does not come from the stramenopiles so the haptophytes appear to have replaced their chloroplasts at least once in their evolutionary history.

The findings show that algal chloroplasts are mosaics, supported by proteins from many different species. This helps us understand why certain species succeed in the wild and how they may respond to environmental changes in the oceans. In the future, these findings may help researchers to engineer new species of algae and plants for food and fuel production.

lineages', consisting of photosynthetic members of the cryptomonads, alveolates (such as dinoflagellates), stramenopiles (also referred to as heterokonts) and haptophytes (*Dorrell and Smith, 2011*; *Baurain et al., 2010*) (see *Table 1* and *Figure 1—figure supplement 1* for definitions). The most prominent of these are the photosynthetic members of the stramenopiles, termed the ochrophytes

**Table 1.** Glossary Box. A schematic figure of eukaryotic taxonomy, showing the evolutionary origins of nuclear and plastid lineages, adapted from previous reviews (*Dorrell and Howe, 2012a*), is shown in *Figure 1—figure supplement 1*.

| | |
|---|---|
| Complex plastids | Plastids acquired through the endosymbiosis of a eukaryotic alga. These include secondary plastids of ultimate red algal origin (such as those found in ochrophytes, haptophytes and cryptomonads), secondary plastids derived from green algae (such as those found in euglenids or chlorarachniophytes), or tertiary plastids such as those found in dinotoms and certain other dinoflagellates (resulting from the endosymbioses of eukaryotic algae that themselves contain plastids of complex origin). |
| CASH lineages | The four major lineages of algae with plastids of secondary or higher red origin, that is to say Cryptomonads, Alveolates (dinoflagellates, and apicomplexans), Stramenopiles, and Haptophytes. |
| Stramenopiles | A diverse and ecologically major component of the eukaryotic tree, containing both photosynthetic members (the ochrophytes), which possess complex plastids of red algal origin, and aplastidic and non-photosynthetic members (e.g. oomycetes, labyrinthulomycetes, and the human pathogen *Blastocystis*), which form the earliest-diverging branches. It is debated when within stramenopile evolution the extant ochrophyte plastid was acquired. |
| Ochrophytes | Photosynthetic and plastid-bearing members of the stramenopiles, including many ecologically important lineages (diatoms, kelps, pelagophytes) and potential model lineages for biofuels research (*Nannochloropsis*). Ochrophytes possess plastids of ultimate red origin, and form the most significant component of eukaryotic marine phytoplankton (*Dorrell and Smith, 2011*; *de Vargas et al., 2015*). |
| Haptophytes | Single-celled, photosynthetic eukaryotes, possessing complex plastids of ultimate red origin. Some haptophytes (the coccolithophorids) are renowned for their ability to form large blooms (visible from space), and to form intricate calcareous shells (*Dorrell and Smith, 2011*; *Bown, 1998*), which if deposited on the ocean floor go on to form a major component of limestone and other sedimentary rocks. |
| HPPG | 'Homologous plastid protein group'. Proteins identified in this study to possess plastid-targeting sequences that are homologous to one another, as defined by BLAST-based HPPG assembly and single gene phylogenetic analysis. |

(*de Vargas et al., 2015*; *Aleoshin et al., 2016*; *Ševčíková et al., 2015*). The ochrophytes include the diatoms, which are major primary producers in the ocean (*Bowler et al., 2010*; *Armbrust et al., 2004*), multicellular kelps, which serve as spawning grounds for marine animals (*Cock et al., 2010*) and the pelagophytes, microscopic algae of which some are known to form harmful blooms (*Gobler et al., 2011*) (*Figure 1*, panel A; *Figure 1—figure supplement 1*). The stramenopiles also contain many aplastidic and non-photosynthetic lineages (e.g., oomycetes), which diverge at the base of the ochrophytes and play important roles as pathogens and in microbial food webs (*Aleoshin et al., 2016*; *Derelle et al., 2016*) (*Figure 1—figure supplement 1*).

Following their acquisition, plastids have undergone a number of evolutionary changes that bound them more intricately with the biology of the host. These include the transfer of plastid-derived genes to the host nucleus (*Dorrell and Howe, 2012a*; *Ruck et al., 2014*; *Stegemann et al., 2003*) and the targeting of proteins encoded within the nucleus to the plastid (*Nowack and Grossman, 2012*; *Kleffmann et al., 2004*). Previous studies have shown that many plastid-targeted proteins are not derived from the genomes of the corresponding endosymbiont lineage (*Curtis et al., 2012*). Proteins encoded by genes acquired from other sources, such as laterally acquired genes (*Qiu et al., 2013*; *Morse et al., 1995*) or previous endosymbiotic organelles historically possessed by the host (*Dorrell and Howe, 2015*, *2012b*), or proteins that have been repurposed from endogenous host organelles (*Fast et al., 2001*; *Harper and Keeling, 2003*) have important roles in supporting the biology of plastid lineages. Other gene transfer events, e.g. from food sources (*Nowack et al., 2016*), bacterial symbionts (*Dunning Hotopp et al., 2007*), viruses (*Gornik et al., 2012*), or diazotrophic non-plastid cyanobacterial endosymbionts (*Prechtl et al., 2004*; *Thompson et al., 2012*) might have also played major roles in the evolution of the diverse range of plastid proteins observed today. It nonetheless remains largely unknown which proteins had the most fundamental roles in establishing current plastid lineages (*Dorrell and Howe, 2012a*), i.e., which plastid proteins represent the ancestral components of plastid-targeted proteomes.

Ochrophytes represent an excellent system in which to reconstruct the origins of plastid proteomes. Firstly, plastid-targeting sequences in different ochrophytes are relatively well conserved, enabling *in silico* prediction of plastid-targeted proteins from a wide range of different species (*Gruber et al., 2015*; *Gschloessl et al., 2008*), in contrast to plastid-targeting sequences within archaeplastid lineages, which are extremely variable (*Fuss et al., 2013*; *Suzuki and Miyagishima, 2010*). Secondly, compared to other CASH lineages (haptophytes, cryptomonads, and dinoflagellates), ochrophytes represent an extremely well characterised system for experimental and bioinformatic investigation, with (to date) eleven complete genomes, and transcriptome libraries available for over 150 species through MMETSP (the Marine Microeukaryote Transcriptome Sequencing Project) and through other sources (*Mock et al., 2017*; *Keeling et al., 2014*). Reliable transformation and other manipulation strategies are also available for multiple species, such as the model diatom *Phaeodactylum tricornutum* (*Siaut et al., 2007*; *Takahashi et al., 2007*; *Radakovits et al., 2013*).

Thirdly, the origin of the ochrophyte plastid is an evolutionarily valuable topic to understand. It is currently not known when the ochrophyte plastid was acquired: whether it originated recently, predates the radiation of aplastidic stramenopile relatives (*Stiller et al., 2014*; *Aleoshin et al., 2016*; *Derelle et al., 2016*), or was acquired prior to the divergence of stramenopiles from their closest relatives, the alveolates (*Janouskovec et al., 2010*). Verifying a late origin for the ochrophyte plastid would thus enable insights into the cellular changes that accompany the transition from a solely heterotrophic to a phototrophic lifestyle (*Aleoshin et al., 2016*; *Derelle et al., 2016*), which is currently not possible for archaeplastids (*Burki et al., 2016*; *Cavalier-Smith et al., 2015*), and difficult for haptophytes and cryptomonads, in which these relatives respectively remain unknown or understudied at a genomic level (*Burki et al., 2016*; *Yabuki et al., 2014*). It has additionally been proposed, based on the presence of large numbers of genes of putative green algal origin in diatom genomes (*Frommolt et al., 2008*; *Petersen et al., 2006*), that the ancestor of ochrophytes once possessed a green algal endosymbiont, which was subsequently replaced via the serial endosymbiosis of a red algal-derived plastid (*Dorrell and Smith, 2011*; *Moustafa et al., 2009*). This hypothesis remains controversial (*Ku et al., 2015*; *Woehle et al., 2011*; *Deschamps and Moreira, 2012*), in particular due to issues associated with the distinction of genes of red and green algal origins in ochrophyte genomes (*Matsuzaki et al., 2004*; *Collén et al., 2013*; *Qiu et al., 2015*). A final evolutionary suggestion regarding ochrophytes is that they have acted as endosymbiotic donors into other CASH lineages. One recent study proposed that haptophytes possess plastids acquired via the

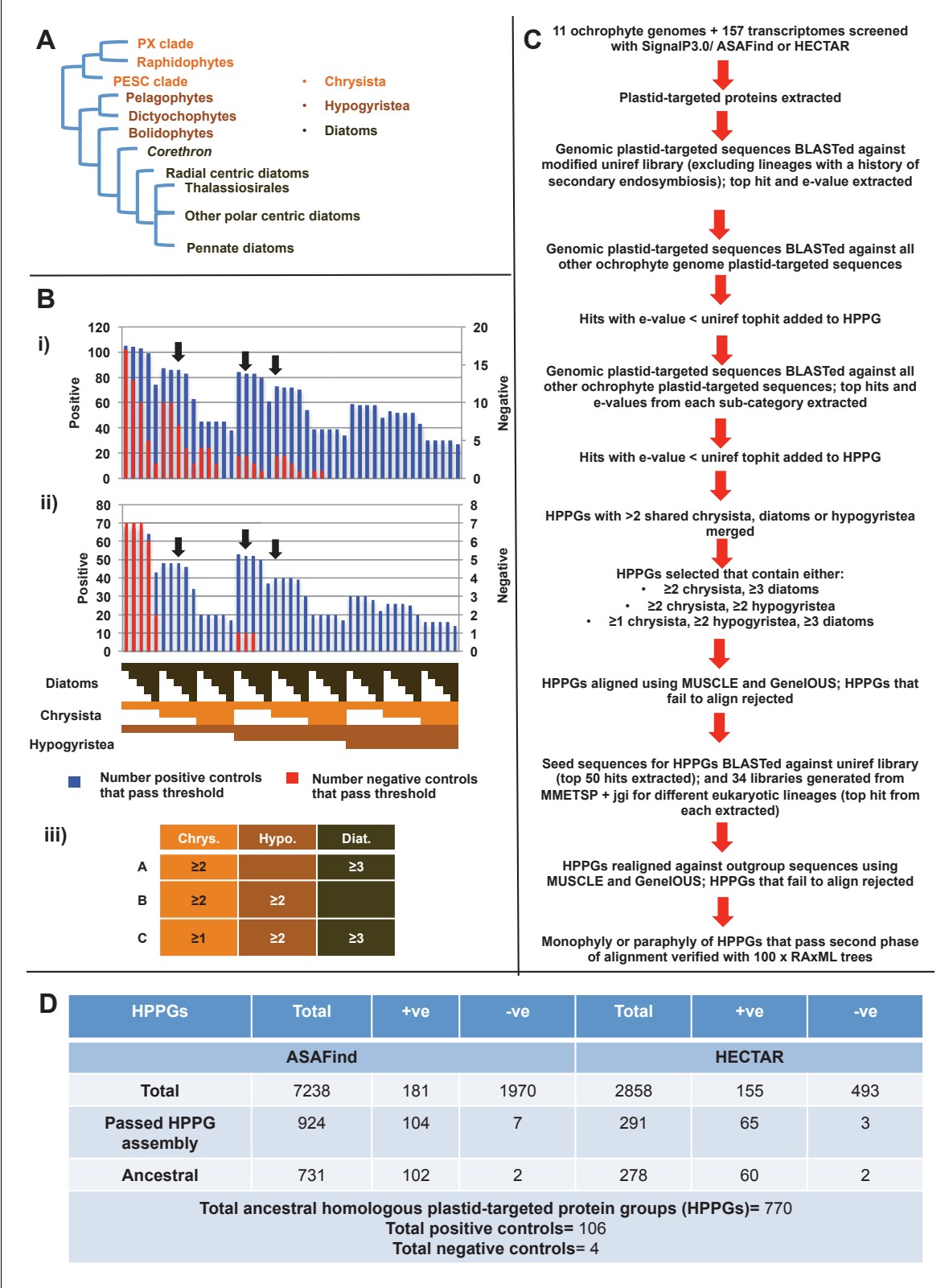

**Figure 1.** Procedure for identification of conserved plastid-targeted proteins in ochrophytes. (Panel **A**) shows a schematic unrooted ochrophyte tree, with the three major ochrophyte lineages (chrysista, hypogyristea, and diatoms) denoted by different coloured labels. 'PX' refers to the combined clade of phaeophytes, xanthophytes and related taxa, and 'PESC' to pinguiophytes, eustigmatophytes, synchromophytes, chrysophytes and relatives. A global overview of the eukaryotic tree of life, including the position of ochrophytes relative to other lineages is shown in *Figure 1—figure supplement*

*Figure 1 continued on next page*

*Figure 1 continued*

*1*. (Panel **B**) shows the number of inferred positive control HPPGs (i.e., HPPGs encoding proteins with experimentally confirmed plastid localisation, or unambiguously plastid function) and negative control HPPGs (i.e., HPPGs encoding proteins with no obvious plastid-targeted orthologues encoded in ochrophyte genomes, but found in haptophyte and cryptomonad genomes) detected as plastid-targeted in different numbers of ochrophyte lineages using ASAFind (i) and HECTAR (ii). The blue bars show the number of positive controls identified to pass a specific conservation threshold, plotted against the left hand vertical axis of the graph, while the red bars show the number of negative controls that pass the same conservation threshold, plotted against the right hand vertical axis of the graph. The number of different sub-categories included in each conservation threshold is shown in a heatmap below the two graphs, with the specific distribution for each bar in the graph shown in the aligned cells directly beneath it. Each shaded cell corresponds to an identified orthologue in one sub-category of a particular ochrophyte lineage: orange cells indicate presence of chrysistan sub-categories; light brown cells the presence of hypogyristean sub-categories; and dark brown cells the presence of diatom sub-categories. In each graph, black arrows label the conservation thresholds inferred to give the strongest separation (as inferred by chi-squared P-value) between positive and negative control sequences. The table (iii) tabulates the three conservation patterns identified as appropriate for distinguishing probable ancestral HPPGs from false positives. (Panel **C**) shows the complete HPPG assembly, alignment and phylogenetic pathway used to identify conserved plastid-targeted proteins. (Panel **D**) tabulates the number of HPPGs built using ASAFind and HECTAR predictions, and the number of non-redundant HPPGs identified in the final dataset. The final total represents the pooled total of non-redundant HPPGs identified with both ASAFind and HECTAR.

The following figure supplement is available for figure 1:

**Figure supplement 1.** Overview of eukaryotic diversity.

endosymbiosis of an ochrophyte (*Stiller et al., 2014*), although the exact identity of this endosymbiotic acquisition remain unresolved. Characterising the ancestral ochrophyte plastid proteome might therefore help answer major questions about the ways in which plastids become established in the host cell, and provide valuable insights into the origins and diversification of other ecologically important algal lineages.

In this study, we present an experimentally verified *in silico* reconstruction of the proteins targeted to the plastid of the last common ochrophyte ancestor. We show that this ancestral plastid-targeted proteome was an evolutionary mosaic, containing 770 proteins from a range of different sources. Our dataset indicates that the ochrophyte plastid was acquired late in stramenopile evolution, following the divergence of extant aplastidic relatives, that plastid-targeted proteins of green algal origin played a significant role in its origin, and that there has been bidirectional integration of the biology of the ochrophyte host and plastid proteomes, such as the ancient recruitment of proteins from both host and endosymbiont to dually support the biology of the plastid and mitochondria. Finally, we show evidence for an ancient endosymbiosis of a specific ochrophyte lineage, an ancestor of the pelagophytes and dictyochophytes, by a common ancestor of the haptophytes, which we propose- based on discrepancies between the origins of the haptophyte plastid proteome and genome- reveals a possible serial endosymbiosis event early in haptophyte evolution, preceding the origins of the current haptophyte plastid. Our work resolves several long-standing questions of ochrophyte evolution, and provides new insights into the origins and diversification of CASH lineages as a whole.

## Results

### In silico reconstruction of an ancestral plastid proteome

We developed an *in silico* pipeline for identifying putatively ancestral plastid-targeted proteins across the ochrophytes (*Figure 1*). We screened a large composite library, comprising eleven different ochrophyte genomes, together with transcriptome data from a further 158 ochrophyte species (Table S1- sheet 1 [*Dorrell et al., 2016*]) using the ochrophyte plastid targeting predictors ASAFind (Table S2- sheet 1 [*Dorrell et al., 2016*]) (*Gruber et al., 2015*) and HECTAR (Table S3- sheet 1 [*Dorrell et al., 2016*]) (*Gschloessl et al., 2008*). Sequences with predicted plastid localisation were binned into eleven taxonomic sub-categories within three major groups (chrysista, hypogyrista, and diatoms) based on recent multigene phylogenies (*Derelle et al., 2016*) (*Figure 1*, panel A; *Figure 1—figure supplement 1*), then assembled by sequence similarity into homologous plastid-targeted protein groups (HPPGs, Materials and methods).

We next tested the level of conservation best able to identify truly ancestral HPPGs. We selected three patterns of conservation that identified the largest number of HPPGs from a positive control dataset of proteins with previously identified plastid-associated functions, and minimised the number identified from a negative control dataset of HPPGs generated using seed sequences from three other published CASH lineage genomes, for which no plastid-targeted orthologues were detected in any ochrophyte genome sequence (Materials and methods; Table S2- sheet 2, sections 1–2; Table S3- sheet 2, sections 1–2 [*Dorrell et al., 2016*]). The selected conservation patterns were: the presence of the protein in a majority ($\geq$2/3) of chrysistan sub-categories and a majority of either diatom ($\geq$3/5) or hypogyristean ($\geq$2/3) sub-categories; or presence in at least one chrysistan sub-category and a majority of both diatoms and hypogyristea (*Figure 1*, panel B). We extracted HPPGs matching the conservation patterns defined above and verified their monophyly within ochrophytes via alignment and single-gene trees (*Figure 1*, panel C; Table S4- sheet 1 [*Dorrell et al., 2016*]). From this, we identified 770 proteins that were probably targeted to the ancestral ochrophyte plastid (*Figure 1*, panel D; Table S4- sheet 2 [*Dorrell et al., 2016*]). This dataset is significantly enriched in proteins from within the positive control dataset and contains significantly fewer proteins from the negative control dataset than would be expected through random assortment (chi-squared test, $p < 1 \times 10^{-10}$; *Figure 1*), confirming its specificity towards probable ancestral plastid-targeted proteins.

## Experimental verification of ancestral ochrophyte HPPGs

We wished to verify that the ancestral ochrophyte plastid-targeted proteins inferred from the *in silico* pipeline are genuinely plastid-targeted. 106 of our inferred ancestral HPPGs include a *P. tricornutum* protein with prior experimental plastid localization, or unambiguous plastid function (*Figure 1*, panel D), but the remainder do not. We selected ten proteins for experimental localisation (*Figure 2*, panel A; Table S5 [*Dorrell et al., 2016*]). These were chosen on the basis of having only non-plastid annotations on the first 50 BLAST hits against the NCBI nr database excluding ochrophytes, hence lack specific *a priori* evidence for a plastid localization. In each case, all of the ochrophyte protein sequences within the alignment had a well conserved central domain, and a highly variable N-terminal domain of between 30 and 50 amino acids containing an ASAFAP motif, consistent with a conserved plastid targeting sequence (*Gruber et al., 2015*) (*Figure 2—figure supplement 1*).

The selected proteins included five aminoacyl-tRNA synthetases that yielded BLAST top hits only against enzymes with cytoplasmic annotations, or of probable prokaryotic origin (*Figure 2—figure supplement 2*). Also included were a GroES-type chaperonin of inferred mitochondrial origin, an Hsp90-type chaperonin of inferred endoplasmic reticulum origin and a pyrophosphate-dependent phosphofructokinase, which is related to cytosolic enzymes from other lineages (*Figure 2—figure supplement 3*), and is distinct from the ATP-dependent phosphofructokinases used by primary plastid lineages (*Smith et al., 2012*). The Mpv17 membrane protein is most closely related to enzymes with peroxisomal functions and localisation (*Wolfe-Simon et al., 2006*; *Gillard et al., 2008*), but lacks any identifiable peroxisomal targeting sequence (PSL, KRR, or a PTS1 motif) (*Ramirez et al., 2014*) in its C-terminus. Finally, a protein ('Novel protein one') that lacks any conserved domains, and yielded no BLAST matches outside of the ochrophytes below an expect value of $1 \times 10^{-05}$ (except for one dinoflagellate sequence), was selected for localisation characterisation (*Figure 2—figure supplement 4*; Table S5 [*Dorrell et al., 2016*]).

We generated C-terminal GFP-fusion constructs for each of these proteins using *P. tricornutum* genes and transformed wild-type *P. tricornutum* (*Figure 2*, panel B; *Figure 2—figure supplement 5*; Table S5 [*Dorrell et al., 2016*]). In each case, we identified GFP fluorescence associated with the plastid. In one case (the peroxisomal membrane protein; *Figure 2*, panel B), the GFP accumulated in a ring around the plastid equator, consistent with a periplastid compartment (PPC) localisation (*Matari and Blair, 2014*; *Tanaka et al., 2015a*). In other cases (such as the five aminoacyl-tRNA synthetases, *Figure 2—figure supplement 5*), the GFP signal localised both within and external to the plastid, consistent with a multipartite localisation within the cell. However, in all cases the proteins tested were at least partially targeted to the plastid.

We additionally generated heterologous GFP fusion constructs for five of the proteins using sequences from the 'dinotom' *Glenodinium foliaceum*, a dinoflagellate alga that harbours permanent endosymbionts of diatom origin (*Dorrell and Howe, 2015*; *Imanian et al., 2010*), and the eustigmatophyte *Nannochloropsis gaditana*, which as a member of the 'PESC clade' is distantly

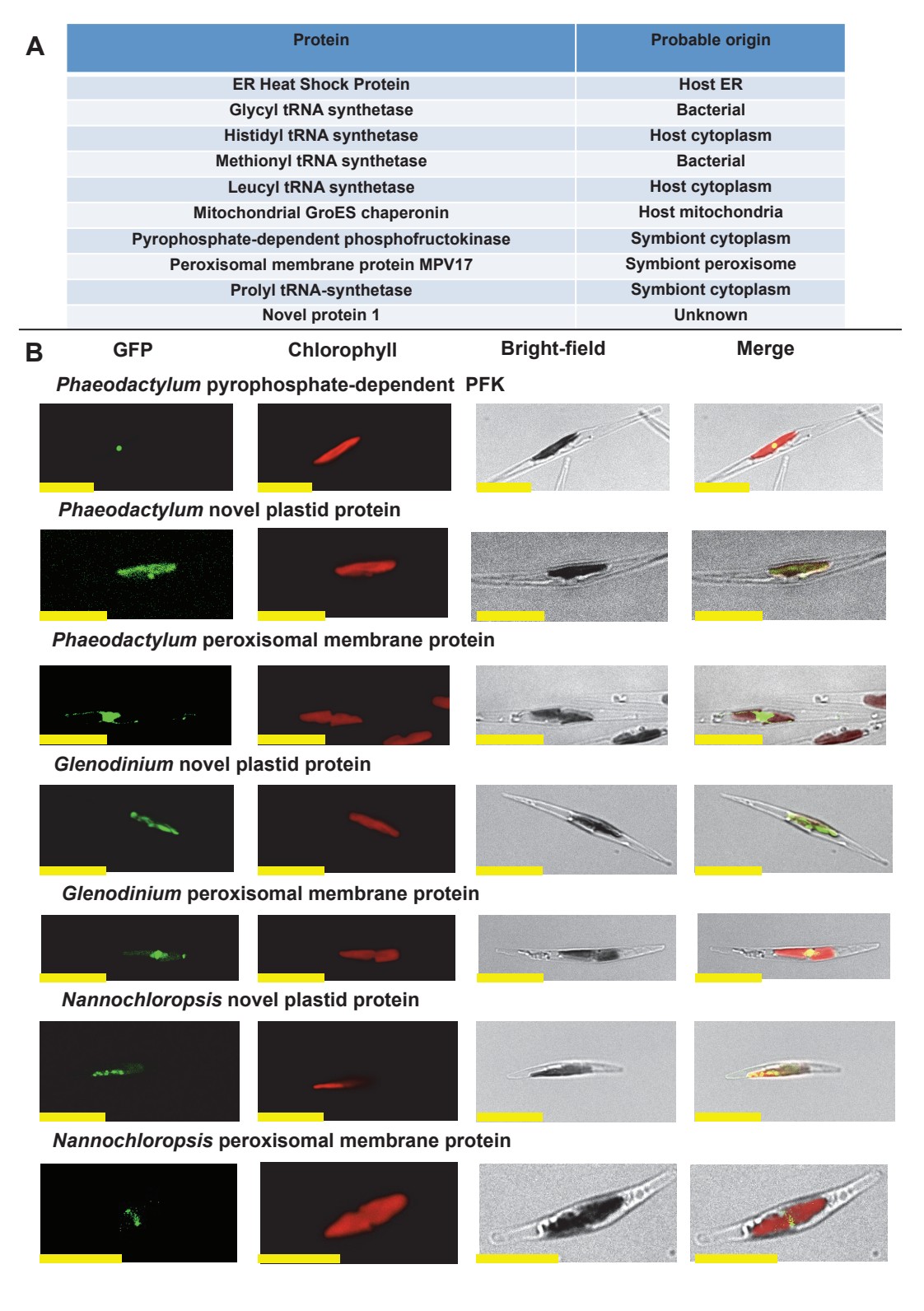

| Protein | Probable origin |
|---|---|
| ER Heat Shock Protein | Host ER |
| Glycyl tRNA synthetase | Bacterial |
| Histidyl tRNA synthetase | Host cytoplasm |
| Methionyl tRNA synthetase | Bacterial |
| Leucyl tRNA synthetase | Host cytoplasm |
| Mitochondrial GroES chaperonin | Host mitochondria |
| Pyrophosphate-dependent phosphofructokinase | Symbiont cytoplasm |
| Peroxisomal membrane protein MPV17 | Symbiont peroxisome |
| Prolyl tRNA-synthetase | Symbiont cytoplasm |
| Novel protein 1 | Unknown |

**B**

GFP / Chlorophyll / Bright-field / Merge

*Phaeodactylum* pyrophosphate-dependent PFK

*Phaeodactylum* novel plastid protein

*Phaeodactylum* peroxisomal membrane protein

*Glenodinium* novel plastid protein

*Glenodinium* peroxisomal membrane protein

*Nannochloropsis* novel plastid protein

*Nannochloropsis* peroxisomal membrane protein

**Figure 2.** Verification of unusual ancestral plastid-targeted proteins. (Panel **A**) lists the ten proteins selected for experimental characterisation and their most probable previous localisation prior to their establishment in the ochrophyte plastid, based on the first 50 nr BLAST hits. Exemplar alignments and single-gene tree topologies for some of these proteins are shown in *Figure 2—figure supplements 1–4*. (Panel **B**) shows the localisation of GFP constructs for copies of two proteins with an unambiguous plastid localisation (a pyrophosphate-dependent PFK, which localises to the pyrenoid, and a *Figure 2 continued on next page*

*Figure 2 continued*

novel plastid protein, with cosmopolitan distribution across the plastid) and one protein with a periplastid localisation (a predicted peroxisomal membrane protein) from the diatom *Phaeodactylum tricornutum,* the diatom endosymbiont of the dinoflagellate *Glenodinium foliaceum* and the eustigmatophyte *Nannochloropsis gaditana,* expressed in *P. tricornutum*. All scale bars = 10 μm. Expression constructs for seven additional *P. tricornutum* proteins and three additional *N. gaditana* proteins with multipartite plastid localisations are shown in *Figure 2—figure supplements 5* and *6*, and control images (wild-type cells, and cells expressing untargeted eGFP) are shown in *Figure 2—figure supplement 7*.

The following figure supplements are available for figure 2:

**Figure supplement 1.** Exemplar ochrophyte plastid protein alignments.

**Figure supplement 2.** Tree of ochrophyte glycyl-tRNA synthetase sequences.

**Figure supplement 3.** Tree of ochrophyte pyrophosphate dependent phosphofructo-1- kinase sequences.

**Figure supplement 4.** Tree of a novel ochrophyte plastid-targeted protein.

**Figure supplement 5.** Multipartite *Phaeodactylum* plastid-targeted proteins.

**Figure supplement 6.** Heterologous expression constructs of multipartite plastid-targeted proteins.

**Figure supplement 7.** Exemplar control images for confocal microscopy.

---

related to *P. tricornutum* on the ochrophyte tree (*Derelle et al., 2016*). We expressed these constructs in *P. tricornutum* (*Figure 2*, panel B; *Figure 2—figure supplement 6*), and, in each case, detected plastid-localized GFP fluorescence similar to the patterns observed with the *P. tricornutum* gene constructs. Overall, our data therefore supports that the ancestral HPPG dataset consists of genuinely conserved plastid-targeted proteins, rather than misidentified proteins of non-plastid function.

## Evolutionary origins of the ochrophyte plastid

### The ochrophyte plastid is an evolutionary mosaic

We wished to identify the evolutionary affinity of each ancestral HPPG in our dataset. In particular, we assessed whether proteins that are of unconventional origin, such as the products of genes endogenous to the host, or genes that have been acquired from other sources such as prokaryotes and green algae, have significantly contributed to the origins of the ochrophyte plastid (*Dorrell and Smith, 2011*; *Moustafa et al., 2009*).

We accordingly determined the closest relative of each ancestral HPPG (Materials and methods). Due to ongoing controversies regarding the evolutionary composition of ochrophyte genomes (*Woehle et al., 2011*; *Deschamps and Moreira, 2012*), we utilised a combined phylogenetic and BLAST top hit approach to robustly infer the most probable origin of each HPPG (Materials and methods; Table S4- sheet 2 [*Dorrell et al., 2016*]). For both the BLAST and phylogenetic analyses, stringent criteria were applied to avoid misidentification due to topological ambiguity, or contamination within individual sequence datasets (*Marron et al., 2016*; *Dorrell et al., 2017*) (Materials and methods). We took the union of these two analyses to produce a dataset of 263 HPPGs for which both phylogenetic and BLAST top hit analyses indicated the same clear evolutionary origin. These origins were grouped into six evolutionary categories, red algae, green algae, aplastidic stramenopiles, other eukaryotes, prokaryotes, and viruses (*Figure 3*, panel A).

Of the 263 HPPGs that were resolved from the combined analysis, 149 (57%) were of red algal, i.e. endosymbiont origin (*Figure 3*, panel A; Table S4- sheet 3 [*Dorrell et al., 2016*]). This is analogous to results from studies of archaeplastid plastid proteomes, in which approximately half of the plastid-targeted proteins are of endosymbiont origin (*Qiu et al., 2013*; *Suzuki and Miyagishima, 2010*). The remaining 114 HPPGs resolved with other sister-groups, consistent with a mosaic origin of the ochrophyte plastid proteome. The most significant of these lineages was green algae (67 HPPGs, 25%), followed by aplastidic stramenopiles (26 HPPGs, 10%), and prokaryotes (21 HPPGs,

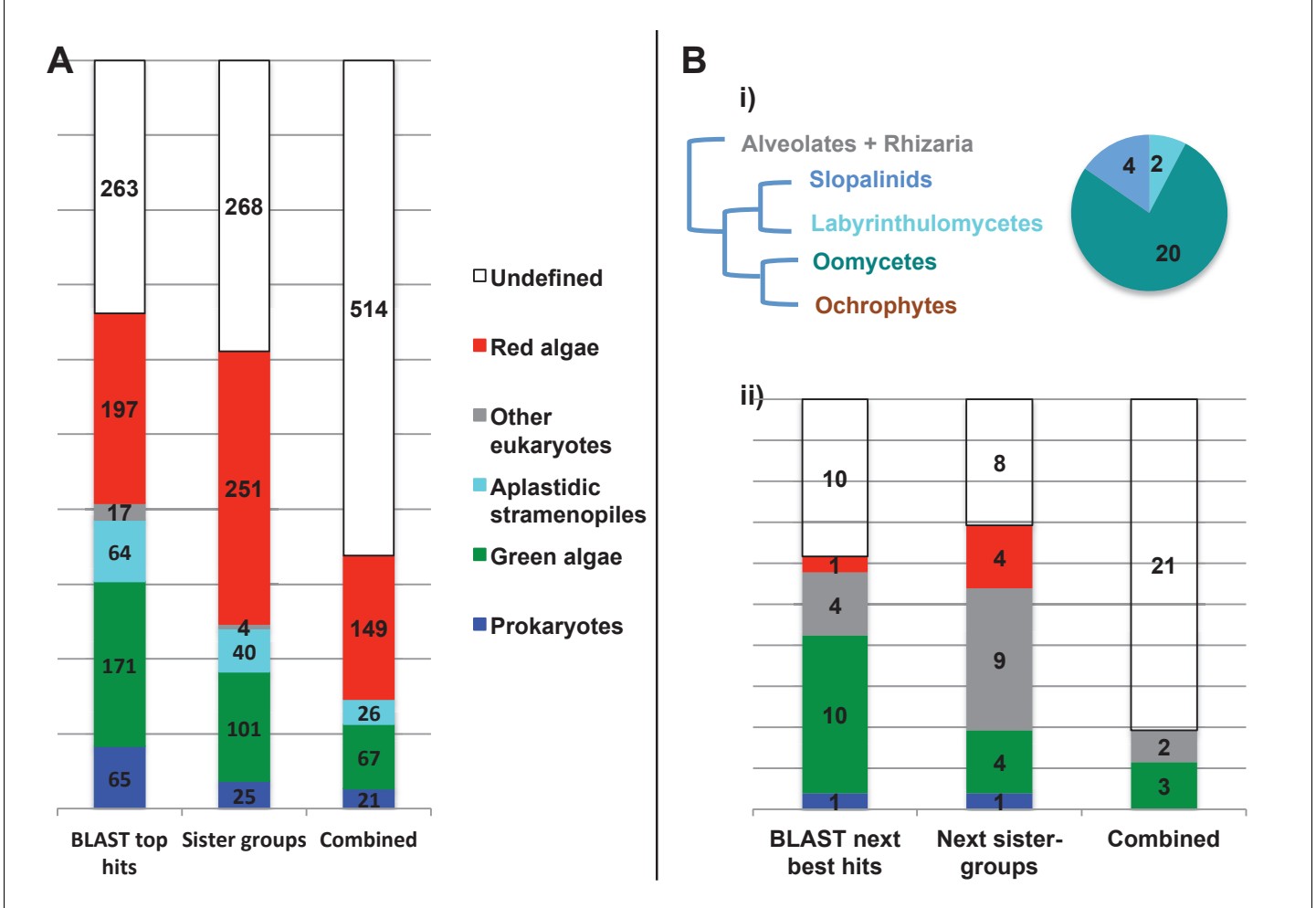

**Figure 3.** Evolutionary origins of the ochrophyte plastid proteome. (Panel **A**) displays the origins inferred by BLAST top hit, phylogenetic analysis, and combined analysis for all ancestral HPPGs. (Panel **B**) shows (i) a schematic diagram of stramenopile taxonomy, with the evolutionary relationships between labyrinthulomycetes, oomycetes, slopalinids and ochrophytes proposed by recent multigene studies (*Derelle et al., 2016*), and the probable closest stramenopile relative (as inferred by BLAST top hit analysis) of the 26 ancestral HPPGs verified by combined analysis to be of aplastidic stramenopile origin, and (ii) the next nearest relative, as inferred through BLAST top hit, phylogenetic and combined analysis, of the 26 aplastidic stramenopile HPPGs verified by combined analysis. The evolutionary categories in this graph are shaded as per in panel A.

8%) (*Figure 3*, panel A). None of the HPPGs were clearly assigned to other eukaryotes or to viruses, consistent with previous assertions that these lineages have contributed very little to ochrophyte evolution (*Bowler et al., 2008*) (*Figure 3*, panel A).

## Late origin of ochrophyte plastids

We wished to determine whether the ochrophyte plastid was acquired by a common ancestor of all stramenopiles or later in ochrophyte evolution. We reasoned that if the ochrophyte plastid was acquired early, i.e., before the divergence of aplastidic relatives, endosymbiotic gene transfer from the red algal symbiont to the host nucleus would have commenced prior to the radiation of the stramenopiles (*Stiller et al., 2009*). Based on the primary evolutionary affinities of each ancestral HPPG (*Figure 3*, panel A), we would expect approximately half of the aplastidic stramenopile-derived proteins to show a deeper red algal origin. We accordingly profiled the deeper evolutionary affinity of each ancestral HPPG of aplastidic stramenopile origin by a combined phylogenetic and BLAST top hit analysis, as before.

First, we noted that the majority (20/26) of the ochrophyte HPPGs with aplastidic stramenopile origins specifically resolved as a sister-group to oomycetes, as opposed to the deeper-branching labyrinthulomycetes or slopalinids (*Figure 3*, panel B; Table S4- sheet 3 [*Dorrell et al., 2016*]). Because oomycetes are the sister-group of ochrophytes (*Aleoshin et al., 2016*; *Derelle et al., 2016*), this suggests that our dataset retains useful phylogenetic signal.

Next, from the 26 ancestral HPPGs of aplastidic stramenopile origin, we identified a clear sister-group to the stramenopile clade for 16 HPPGs using BLAST, and for 18 HPPGs using single-gene trees (*Figure 3*, panel B). However, only one BLAST top hit and four trees showed a deeper red algal affinity (*Figure 3*, panel B). These proportions are significantly smaller than the proportions of ochrophyte proteins of red origin in the entire ancestral HPPG dataset (expected frequencies: 9.54 BLAST top hits, 10.7 sister-groups; chi-squared test, p≤0.01; *Figure 3*, panels A, B). In five cases we identified the same deeper affinity through combined BLAST top hit and tree sister-group analysis, but none of these were of red algal origin (*Figure 3*, panel B). We conclude that plastid-targeted proteins in ochrophytes that are related to aplastidic stramenopile proteins are predominantly not of red origin. This is consistent with a late origin for the ochrophyte plastid, following the divergence of the ochrophytes and oomycetes.

## A significant green algal contribution to ochrophyte plastid evolution

Previous reports of green genes in ochrophyte genomes have been controversial due to a paucity of red algal sequence data (*Moustafa et al., 2009*; *Deschamps and Moreira, 2012*; *Bowler et al., 2008*). We were able to avail in our pipeline of sequence information from five complete red algal genomes (*Matsuzaki et al., 2004*; *Collén et al., 2013*; *Nakamura et al., 2013*; *Bhattacharya et al., 2013*; *Schönknecht et al., 2013*) and twelve red algal transcriptomes (*Keeling et al., 2014*; *Matasci et al., 2014*), allowing us to more clearly infer the reliability of the green signal in ochrophytes. We tested whether the inferred green algal origin could be due to a protein family's absence from red algal lineages (*Figure 4*, panel A). For the majority of our green HPPGs (40/67), an orthologue was identified in at least four of the five major red algal sub-categories considered (cyanidiales, bangophytes and florideophytes, compsopogonophytes and stylonematophytes, porphyridiophytes, and rhodellophytes; *Figure 4*, panel B; *Figure 4—figure supplement 1*; Table S4- sheet 4 [*Dorrell et al., 2016*]). We therefore conclude that these green genes were not misidentified as the result of undersampling within red sequence libraries, or secondary gene loss events in the red algae (*Ku et al., 2015*; *Qiu et al., 2015*).

We then considered whether the green genes in our dataset originate from a specific source within the green algae. Phylogenetic analyses of the HPPGs of verified green origin exhibited a strong bias toward chlorophyte origins. Ochrophytes branched as sister-groups to individual or multiple chlorophyte lineages in 51 of the 67 trees (*Figure 4*, panel C; *Figure 4—figure supplement 2*). Similarly, we noted a strong predominance of chlorophyte lineages amongst BLAST top hits (56/67) despite the fact that these lineages only correspond to approximately 25% of the green sequences present in our libraries (*Figure 4—figure supplement 3*; Table S4- sheet 3 [*Dorrell et al., 2016*]). In contrast, only 16 of the single-gene trees for HPPGs of verified green origin recovered a sister-group relationship between ochrophytes and all green lineages (chlorophytes and streptophytes), none recovered a specific sister-group relationship between ochrophytes and streptophytes (*Figure 4*, panel C), and only 11 of the BLAST top hits were to streptophyte sequences (*Figure 4—figure supplement 2*; Table S4- sheet 3 [*Dorrell et al., 2016*]). This bias is inconsistent with the green ancestral HPPGs being of misidentified red origin, or originating at a deeper position within the green algae, in which case they should show a more stochastic distribution of evolutionary affinities across all green lineages (*Woehle et al., 2011*).

Next, we tested whether our data supported a single origin for the green genes within the chlorophytes, or whether the HPPGs of green origin arose through gene transfer events from multiple chlorophyte lineages. We identified all amino acids that were uniquely shared between ochrophytes and chlorophytes in the 31 green HPPGs for which we found no evidence of gene duplication or subsequent lateral gene transfer into green algae, ochrophytes, or other major photosynthetic eukaryotes (Table S6- sheets 1, 2 (*Dorrell et al., 2016*); Materials and methods). We then inferred the most probable origin in the green algal tree for each uniquely shared residue as well as the earliest possible origin, taking into account gapped and missing positions (*Figure 4*, panel D; *Figure 4—*

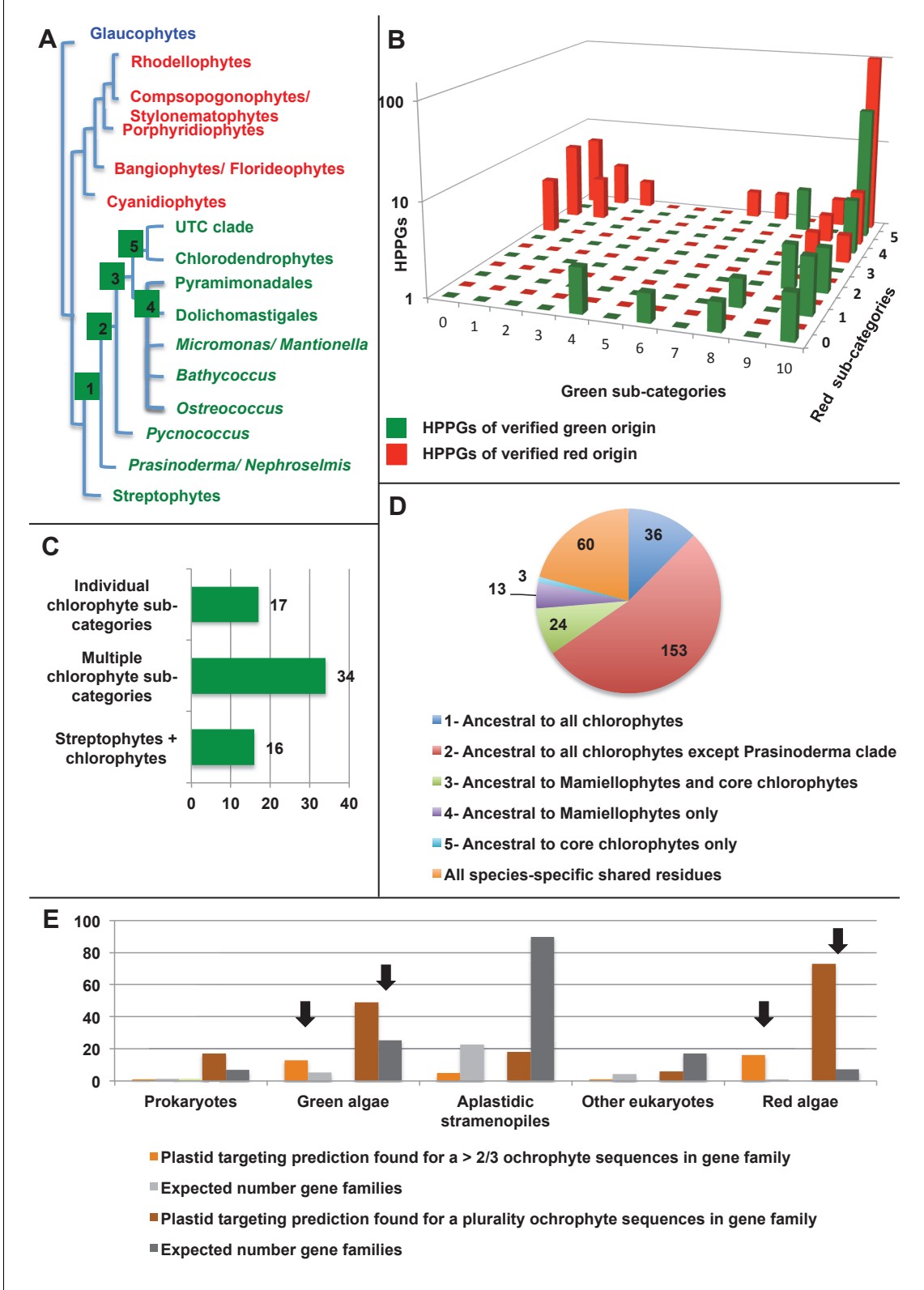

**Figure 4.** Verification and origins of the green signal in ochrophyte plastids. (Panel **A**) shows a schematic tree of the 11 archaeplastid sub-categories with which each green HPPG alignment was enriched prior to phylogenetic analysis. The topology of the red and green algae are shown according to previously published phylogenies (*Leliaert et al., 2011*; *Yoon et al., 2006*). Green sub-categories are in green text; red algal sub-categories in red text; and other sub-categories are in blue text. Five ancestral positions within the green algal tree inspected in subsequent analyses are labelled with

*Figure 4 continued on next page*

*Figure 4 continued*

coloured boxes. (Panel **B**) shows the number of HPPGs of verified red (red bars) or green origin (green bars) for which orthologues were identified in different numbers green sub-categories (plotted on the x-axis) and red sub-categories (plotted on the z-axis). An equivalent graph showing only HPPGs for which a glaucophyte orthologue was detected is shown in *Figure 4—figure supplement 1*. (Panel **C**) compares the number of trees in which HPPGs of verified green origin resolve as a sister group to all green lineages (including chlorophytes and streptophytes); to multiple chlorophyte sub-categories but to the exclusion of streptophytes; and to individual chlorophyte sub-categories only. A detailed heatmap of the evolutionary distribution of the green sub-categories detected in each sister-group is shown in *Figure 4—figure supplement 2*, and the distribution of BLAST top hits within each sub-category is shown in *Figure 4—figure supplement 3*. (Panel **D**) lists the number of residues inferred from a dataset of 32 ochrophyte HPPGs of verified green origin, which have been subsequently entirely vertically inherited in all major photosynthetic eukaryotic lineages, to be uniquely shared between ochrophytes and some but not all green lineages, hence might represent specific synapomorphic residues. Residues are categorized by inferred origin point within the tree topology shown in panel A, i.e., each of the five ancestral nodes labelled. A final category shows all of the residues inferred to be specifically shared with one green sub-category, and not with any other. The distribution of residues based on the earliest possible origin point (taking into account gapped and missing residues in each HPPG alignment) is shown in *Figure 4—figure supplement 4*. (Panel **E**) shows the number of the 7140 conserved gene families inferred to have been present in the last common ochrophyte ancestor that are predicted by ASAFind to encode proteins targeted to the plastid, subdivided by probable evolutionary origin, and the number expected to be present in each category assuming a random distribution of plastid-targeted proteins across the entire dataset, independent of evolutionary origin. Evolutionary categories of proteins found to be significantly more likely (chi-squared test, p=0.05) to encode plastid-targeted proteins than would be expected are labelled with black arrows. An equivalent distribution of plastid-targeted proteins inferred using HECTAR is shown in *Figure 4—figure supplement 5*.

The following figure supplements are available for figure 4:

**Figure supplement 1.** Sampling richness associated with ancestral HPPGs of green algal origin.

**Figure supplement 2.** Heatmaps of nearest sister-groups of ancestral HPPGs of verified green origin.

**Figure supplement 3.** Specific origins of green HPPGs as inferred from BLAST top hit analyses.

**Figure supplement 4.** Earliest evolutionary origins of shared plastid residues.

**Figure supplement 5.** Origins and HECTAR based targeting tests of proteins encoded by conserved ochrophyte gene clusters.

*figure supplement 4*; Table S7- sheets 1, 3 [*Dorrell et al., 2016*]). In both analyses the majority of the uniquely shared residues were inferred to have originated in a common ancestor of all chlorophytes, or of all chlorophyte lineages excluding the basal *Prasinoderma/ Nephroselmis* sub-category (189/289 positions in observed analysis; 100/147 positions in the earliest possible analysis; *Figure 4*, panel D; *Figure 4—figure supplement 4*; Table S7- sheets 1, 3 [*Dorrell et al., 2016*]). All other nodes within the green tree, including all specific green sub-categories, shared much smaller numbers of residues with ochrophytes (*Figure 4*, panel D; *Figure 4—figure supplement 4*; Table S7- sheets 1, 3 [*Dorrell et al., 2016*]). Thus, our data is congruent with the majority of the ochrophyte green genes originating from deep within the chlorophyte lineage.

Finally, we considered whether the green genes that function in ochrophyte plastids were more likely to have been acquired through endosymbiosis, or through lateral gene transfers, for example from a food organism (*Keeling and Palmer, 2008*; *Doolittle, 1998*) or other intracellular symbiont (*Dorrell and Howe, 2012a*). We reasoned that if the green genes in ochrophytes were predominantly of endosymbiotic origin, they should encode more plastid-targeted proteins than genes of alternative origin, in the same manner as genes of cyanobacterial origin retained in archaeplastid genomes are biased towards encoding proteins with plastid functions (*Dorrell and Howe, 2015*). We accordingly constructed a secondary dataset, consisting of 7140 non-redundant gene families that are broadly distributed across the ochrophytes, and tested the targeting preferences of proteins from each HPPG (*Figure 4*, panel E; *Figure 4—figure supplement 5*; Table S8- sheet 1 [*Dorrell et al., 2016*]). 871 gene families resolved with the green algae per BLAST top hit analysis (*Figure 4—figure supplement 5*; Table S8- sheet 2 [*Dorrell et al., 2016*]). Using both ASAFind (*Gruber et al., 2015*) and HECTAR (*Gschloessl et al., 2008*), gene families of predicted green algal origin were significantly more likely to encode proteins with plastid-targeting predictions than the dataset as a whole (chi-squared, $p<1E^{-03}$; *Figure 4*, panel E; *Figure 4—figure supplement 5*; Table S8- sheet 3 [*Dorrell et al., 2016*]). We also observed a similar, though stronger, bias towards

plastid-targeted proteins among the proteins of red algal origin (chi-squared, $p<1E^{-40}$; *Figure 4*, panel E; *Figure 4—figure supplement 5*; Table S8- sheet 3 [*Dorrell et al., 2016*]). Collectively, our data support the presence of genes of chlorophyte origin in the last common ochrophyte ancestor, which are biased towards encoding proteins with predicted plastid localisations, consistent with an acquisition through a plastid endosymbiosis event.

## Functional consequences of mosaic origins for the ochrophyte plastid

### Metabolic completeness of the ochrophyte plastid

We identified effectively complete core plastid metabolism pathways within the ancestral HPPG dataset (*Figure 5*, panel A; *Figure 5—figure supplement 1*; Table S9- sheet 1 [*Dorrell et al., 2016*]). The majority of the remaining proteins remain plastid-encoded in some ochrophyte lineages, or are dispensable for the metabolic pathway (*Figure 5—figure supplements 1* and *2*) (*Ershov et al., 2000*; *Rohdich et al., 2002*; *Gutierrez-Marcos et al., 1996*). In four cases (isopropylmalate synthase, sedoheptulose bisphosphatase, 3-dehydroquinate synthase, and shikimate kinase) lateral gene transfer and replacement events have occurred into individual ochrophyte lineages since their radiation, preventing identification of a single HPPG within the ancestral dataset (*Figure 5*, panel A; *Figure 5—figure supplements 2–6*). Taking these exceptions into account, we conclude that the ancestral ochrophyte plastid proteome contained the fundamental components of core plastid metabolism.

### Mosaic origins of ochrophyte plastid metabolism

Given the mosaic evolutionary origins of ancestral ochrophyte plastid-targeted proteins, we wondered whether certain evolutionary affinities might correlate with specific metabolic functions. It has previously been speculated, for example, that genes acquired by diatoms from green algae might have a specific role in tolerating variable light regimes (*Frommolt et al., 2008*; *Dittami et al., 2010*; *Coesel et al., 2008*) or eliminating toxic substances from diatom plastids (*Chan et al., 2011*). We noted that many of the pathways in the ochrophyte plastid utilise a mixture of genes of red, green, host and prokaryotic origin (*Figure 5—figure supplement 1*), which would suggest a converse scenario: that the mosaic origins of the ochrophyte plastid have led to the functional mixing of enzymes with disparate evolutionary origins.

Consistent with this latter idea, we found very little evidence that individual categories of HPPG (i.e., red algal, green algal, prokaryotic or host origin) are associated with particular KOG annotations, as inferred by chi-squared testing ($p<0.05$) against a null hypothesis that all KOG families and classes are homogenously distributed across the ancestral HPPG dataset, independent of evolutionary origin (*Figure 5*, panel B; *Figure 5—figure supplement 7*; Table S9- sheet 2 [*Dorrell et al., 2016*]). The notable exceptions are prokaryotic HPPGs being elevated in information storage and processing proteins, particularly those involved in translation, while HPPGs of host origin were enriched in proteins involved in cellular processes and signalling relative to the ancestral HPPG set as a whole (*Figure 5*, panel B; *Figure 5—figure supplement 7*; Table S9- sheet 2 [*Dorrell et al., 2016*]). In contrast, several KOG categories were more highly represented in the ancestral HPPG set than in HPPGs as a whole (*Figure 5*, panel B; *Figure 5—figure supplement 7*; Table S9- sheet 2 [*Dorrell et al., 2016*]).

A related question is whether proteins that catalyse adjacent steps of a biochemical pathway tend to have shared or different evolutionary affinities. Multiple sets of non-native proteins might be preferentially utilised by ochrophyte plastids due to performing concerted steps in individual metabolic pathways or cellular processes (*Dorrell and Smith, 2011*; *Frommolt et al., 2008*; *Yurchenko et al., 2016*). In this instance, pairs of proteins that interact with one another would be more likely to come from the same evolutionary origin than would be expected by random association. Alternatively, early ochrophyte plastids might have had no preference for utilising interacting proteins of the same evolutionary origin, in which case proteins involved in specific metabolic pathways might frequently have different evolutionary origins to adjacent enzymes in the same pathway. Of the 313 pairs of such biochemical neighbours identified in the ancestral HPPGs, only 44 shared the same evolutionary origin, which is no different than that which would be expected by chance (expected number 41.05; chi-squared test, $p=0.541$; *Figure 5*, panel C; Table S9- sheet 3

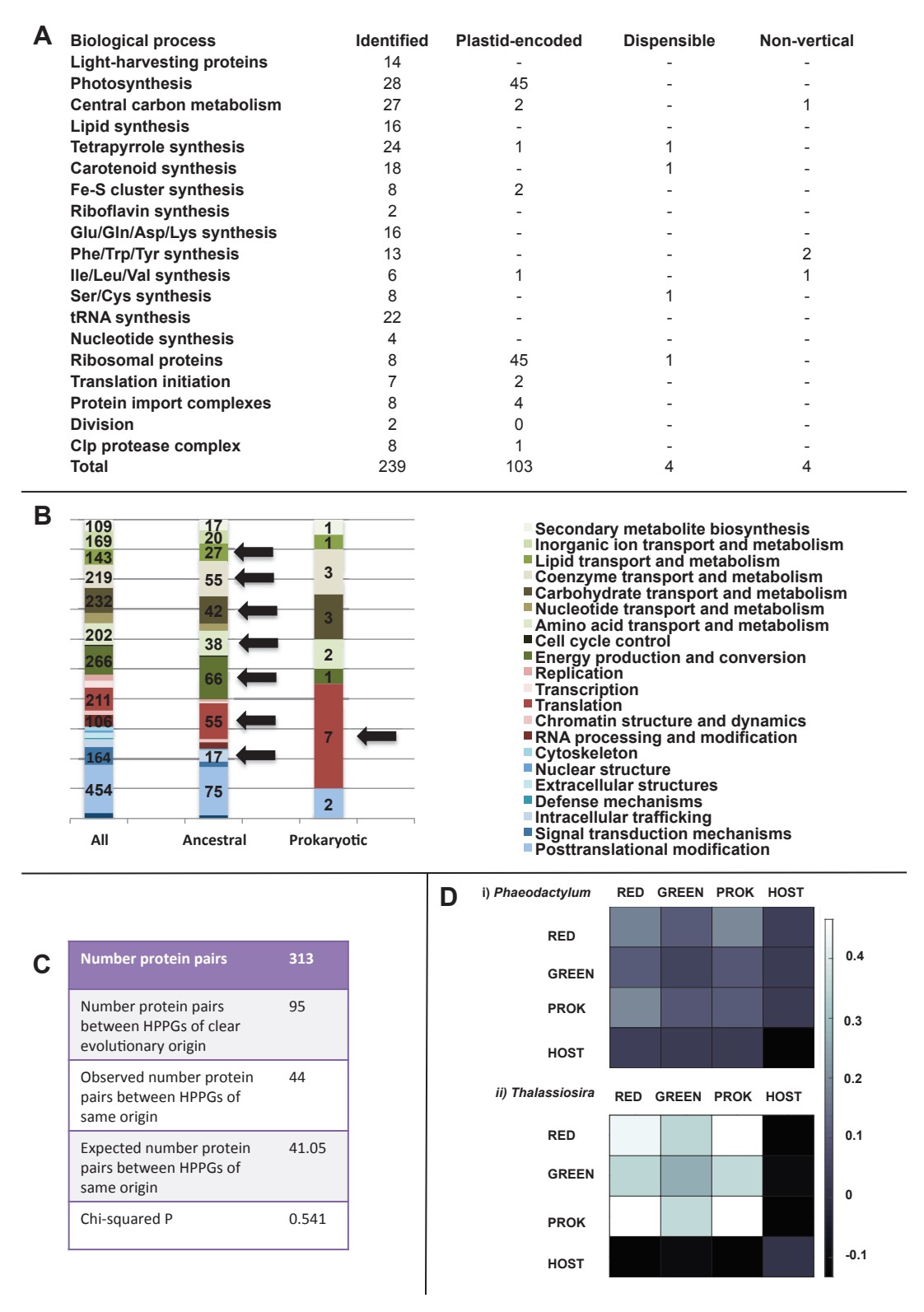

**A**

| Biological process | Identified | Plastid-encoded | Dispensable | Non-vertical |
|---|---|---|---|---|
| Light-harvesting proteins | 14 | - | - | - |
| Photosynthesis | 28 | 45 | - | - |
| Central carbon metabolism | 27 | 2 | - | 1 |
| Lipid synthesis | 16 | - | - | - |
| Tetrapyrrole synthesis | 24 | 1 | 1 | - |
| Carotenoid synthesis | 18 | - | 1 | - |
| Fe-S cluster synthesis | 8 | 2 | - | - |
| Riboflavin synthesis | 2 | - | - | - |
| Glu/Gln/Asp/Lys synthesis | 16 | - | - | - |
| Phe/Trp/Tyr synthesis | 13 | - | - | 2 |
| Ile/Leu/Val synthesis | 6 | 1 | - | 1 |
| Ser/Cys synthesis | 8 | - | 1 | - |
| tRNA synthesis | 22 | - | - | - |
| Nucleotide synthesis | 4 | - | - | - |
| Ribosomal proteins | 8 | 45 | 1 | - |
| Translation initiation | 7 | 2 | - | - |
| Protein import complexes | 8 | 4 | - | - |
| Division | 2 | 0 | - | - |
| Clp protease complex | 8 | 1 | - | - |
| Total | 239 | 103 | 4 | 4 |

**B**

Secondary metabolite biosynthesis
Inorganic ion transport and metabolism
Lipid transport and metabolism
Coenzyme transport and metabolism
Carbohydrate transport and metabolism
Nucleotide transport and metabolism
Amino acid transport and metabolism
Cell cycle control
Energy production and conversion
Replication
Transcription
Translation
Chromatin structure and dynamics
RNA processing and modification
Cytoskeleton
Nuclear structure
Extracellular structures
Defense mechanisms
Intracellular trafficking
Signal transduction mechanisms
Posttranslational modification

**C**

| Number protein pairs | 313 |
|---|---|
| Number protein pairs between HPPGs of clear evolutionary origin | 95 |
| Observed number protein pairs between HPPGs of same origin | 44 |
| Expected number protein pairs between HPPGs of same origin | 41.05 |
| Chi-squared P | 0.541 |

**D**

i) *Phaeodactylum*

ii) *Thalassiosira*

**Figure 5.** Functional mixing of the ancestral ochrophyte HPPGs. (Panel **A**) tabulates nineteen different fundamental plastid metabolism pathways and biological processes recovered in the ancestral HPPG dataset. Detailed information concerning the origin and identity of each component of each pathway is provided in *Figure 5—figure supplement 1*, and an overview and phylogenetic trees of each of the non-vertically inherited enzymes identified are provided in *Figure 5—figure supplements 2–6*. (Panel **B**) compares the distribution of individual KOG families in the complete HPPG

*Figure 5 continued*

library, the ancestral HPPG dataset, and HPPGs of verified prokaryotic origin. KOG families pertaining to metabolism are shown in shades of green, families pertaining to information storage are shown in shades of red, and families pertaining to cellular processes are shown in shades of blue. Families with unknown KOG classification or general function predictions only are not shown. KOG classes that are enriched in the ancestral HPPG dataset compared to the relative proportions of each KOG class found in the full HPPG dataset, or in individual ancestral HPPGs of prokaryotic origin compared to the ancestral HPPG dataset (as inferred by chi-squared test, p<0.05), are labelled with black horizontal arrows. No such enrichments were observed in any evolutionary category of ancestral HPPGs other than prokaryotes, hence analogous distributions of HPPGs of red algal, green algal and host origin are not shown. Overviews of the broader KOG classes that are enriched either in the ancestral HPPG dataset, or in specific evolutionary categories of ancestral HPPG, are shown in *Figure 5—figure supplement 7*. (Panel **C**) tabulates the number of ancestral HPPGs performing consecutive metabolic functions, or that are likely to have direct regulatory interactions, alongside the number of these protein pairs in which both members are of verified evolutionary origin; the number observed where both members possess the same evolutionary origin; the expected number of protein pairs where both members possess the same evolutionary origin; and the chi-squared probability of similarity between the observed and expected values. (Panel **D**) shows heatmaps for the pairwise correlation coefficients of expression for genes encoding different evolutionary categories, as verified using combined BLAST top hit and single-gene tree analysis, of ancestral HPPGs in the model diatoms *Phaeodactylum tricornutum* (i) and *Thalassiosira pseudonana* (ii). A scale bar showing the relationship between shading and correlation coefficient is shown to the right of the heatmaps. Boxplots comparing the individual expression profiles of different categories of ancestral HPPG, and the associated ANOVA P values calculated, are shown in *Figure 5—figure supplement 8* (for *P. tricornutum*) and *Figure 5—figure supplement 9* (for *T. pseudonana*).

The following figure supplements are available for figure 5:

**Figure supplement 1.** Reconstructed metabolism pathways and core biological processes in the ancestral ochrophyte plastid.

**Figure supplement 2.** Core plastid metabolism proteins not identified within the ancestral HPPG dataset.

**Figure supplement 3.** Tree of ochrophyte sedoheptulose- 7-bisphosphatase sequences.

**Figure supplement 4.** Tree of ochrophyte 3-dehydroquinate synthase sequences.

**Figure supplement 5.** Tree of ochrophyte isopropylmalate dehydrogenase sequences.

**Figure supplement 6.** Tree of ochrophyte shikimate kinase sequences.

**Figure supplement 7.** KOG classes associated with different categories of HPPGs.

**Figure supplement 8.** Coregulation of genes incorporated into HPPGs of different origin in the model diatom *Phaeodactylum tricornutum*.

**Figure supplement 9.** Coregulation of genes incorporated into HPPGs of different origin in the model diatom *Thalassiosira pseudonana*.

[*Dorrell et al., 2016*]), Thus, interactions between proteins of different evolutionary origin were forged early in the evolution of the ochrophyte plastid.

Finally, we sought correlations between expression dynamics and evolutionary affinity, taking advantage of microarray data from *P. tricornutum* and *T. pseudonana* (*Ashworth et al., 2016*) (Table S10- sheets 1–4 [*Dorrell et al., 2016*]). We found no evidence that ancestral HPPG genes of any evolutionary origin had more similar expression dynamics to each other than to those of other evolutionary origins (ANOVA, p≤0.05; *Figure 5*, panel D; *Figure 5—figure supplements 8* and *9*; Table S10-sheet 5 [*Dorrell et al., 2016*]). For example, in both species, genes of green origin show a weaker average positive coregulation with one another than they do to genes from the same species of red or of prokaryotic origin (*Figure 5*, panel D). Thus, the chimeric origins of the ochrophyte plastid has enabled extraordinary functional mixing of proteins from early in its evolution, with each of the different donors contributing proteins with a broad range of biochemical functions and transcriptional patterns in response to changing physiological conditions.

## Ancient origins of chimeric plastid-targeted proteins

We considered whether the mixing of proteins from different evolutionary sources might have more substantially changed the biology of the ochrophyte plastid. It has recently been reported (*Méheust et al., 2016*) that proteins of chimeric evolutionary origin, generated by the fusion of

domains from different evolutionary sources, form a significant component of plastid proteomes. Thus, the chimeric origins of the ochrophyte plastid might have enabled the creation of syncretic proteins not found in the endosymbiont or host ancestors. We identified orthologues of seven chimeric proteins identified in this study within our dataset, underlining their importance for the establishment of the ochrophyte plastid (*Figure 6*, panel A) (*Méheust et al., 2016*).

Next, we assessed whether the mosaic composition of the ochrophyte plastid proteome had also enabled the establishment of novel chimeric fusion proteins, unique to ochrophyte plastids. Using the taxonomic subdivisions erected for this study, we identified further chimerism events in members of 42 ancestral HPPGs (*Figure 6*, panel B; Table S9- sheet 1, sections 4, 5; Table S11 [*Dorrell et al., 2016*]). These include three HPPGs (e.g. NADH-ubiquinone dehydrogenase) in which chimeric proteins have formed through the fusion of modules of prokaryotic origin to others of eukaryotic origin, and seven HPPGs (e.g. translation factor EF-3b, and an N6-adenine DNA methyltransferase) in which fusion events have occurred between modules of red origin and modules of green origin (*Figure 6*, panel B). To our knowledge, neither of these types of fusion event have previously been reported for plastid-targeted proteins (*Méheust et al., 2016*). Twenty of the chimeric HPPGs (47.6%) contain a domain of inferred green origin and 18 (43.8%) contain a domain of host origin.

Amongst the chimeric proteins identified, we found two that probably fused in the ochrophyte ancestor (*Figure 6*, panels A, B). In one case, a bifunctional protein containing an N-terminal 3,4-dihydroxy-2-butanone 4-phosphate (DHBP) synthase and C-terminal GTP cyclohydrolase II protein, which performs two consecutive steps of riboflavin biosynthesis (*Herz et al., 2000*), has formed through the fusion of a cyclohydrolase domain of probable host origin to a synthase domain of probable red algal or actinobacterial origin (*Figure 6—figure supplements 1* and *2*). While bifunctional DHBP synthase/ GTP cyclohydrolase proteins are known in bacteria, red algae and plants (*Figure 6—figure supplement 1*) (*Matsuzaki et al., 2004*; *Herz et al., 2000*), in these taxa the DHBP synthase domain is located at the protein C-terminus; thus, an analogous but topographically distinct fusion protein has evolved in ochrophytes. In a second, previously reported case (*Méheust et al., 2016*), a C-terminal plastid-targeted Tic20 subunit of red algal origin has become fused to an N-terminal EF-hand motif, for which no clear evolutionary outgroup (to an e value of below $1 \times 10^{-05}$) could be found (*Figure 6—figure supplement 3*). Thus, the fusion of proteins of different evolutionary origins has generated new functions in the ochrophyte plastid proteome.

## Ancestral and bidirectional origins of dual-targeting in ochrophytes

Finally, we considered whether the acquisition of the ochrophyte plastid might have also fundamentally altered the biology of the host cell, by contributing proteins to host processes and structures outside the plastid. As an exemplar system, we considered dual-targeting of proteins to plastids and mitochondria, which is known to occur extensively in plants (*Xu et al., 2013*; *Duchêne et al., 2005*), and has recently been documented in diatoms (*Gile et al., 2015*) and in other complex plastid lineages (*Gile et al., 2015*; *Hirakawa et al., 2012*). Previous studies have speculated that dual-targeting may arise early in plastid evolution, for example through the retargeting of proteins from the host mitochondria to the plastid, or equally via the adaptation of proteins of plastid origin to the mitochondria (*Qiu et al., 2013*; *Xu et al., 2013*).

We indeed identified proteins that appeared to be dual-targeted to the plastid and a secondary organelle (*Figure 2—figure supplements 5* and *6*), which we verified to be the mitochondria using Mitotracker orange (*Figure 7* panel A). In at least two cases (histidyl- and prolyl-tRNA synthetase) this dual-targeting is a conserved feature, as we identified the same fluorescence patterns both in *P. tricornutum* and using heterologous expression constructs from *G. foliaceum* and *N. gaditana* (*Figure 7*, panel A; *Figure 7—figure supplement 1*). To determine whether dual-targeted proteins were ancestrally present in the ochrophyte plastid, we developed an *in silico* pipeline, based on experimental data, to identify probable dual-targeted proteins from within the HPPG dataset (*Figure 7—figure supplement 2*; Table S12- sheet 1 [*Dorrell et al., 2016*]). In total, we identified 1103 HPPGs that included at least one member that was probably dual-targeted to plastids and mitochondria (Table S12- sheet 1 [*Dorrell et al., 2016*]). 34 of these HPPGs passed the conservation thresholds previously inferred to signify an ancestral origin (Table S12- sheet 1 [*Dorrell et al., 2016*]). Thus, dual-targeting is an ancestral feature of the ochrophyte plastid.

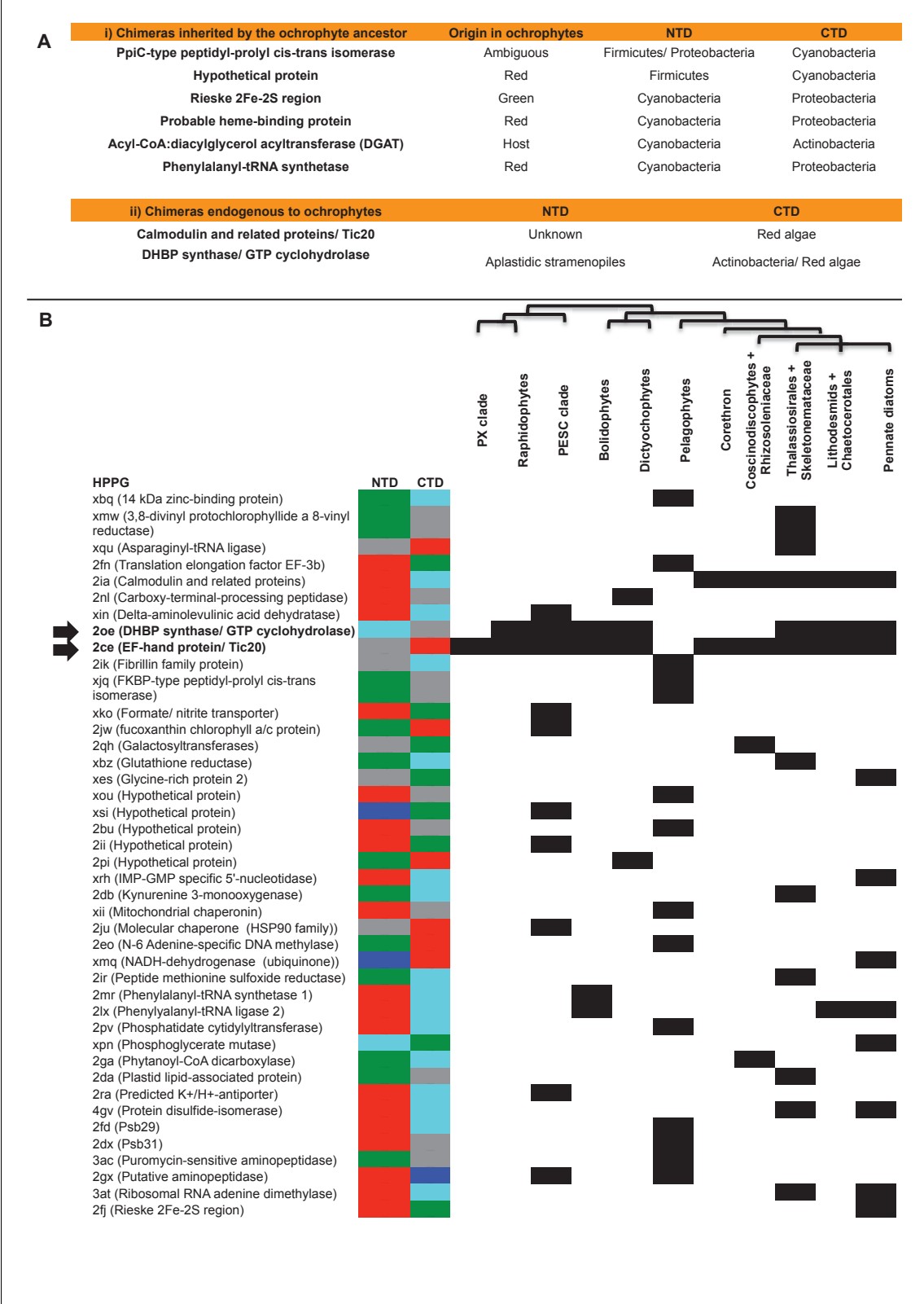

**Figure 6.** Origins of chimeric proteins in the ochrophyte plastid. (Panel **A**) tabulates eight ancestral HPPGs containing domains of cyanobacterial and non-cyanobacterial origin, as previously identified (*Méheust et al., 2016*) that were inherited by the ochrophyte plastid, and two chimeric ancestral HPPGs which are probably of specific ochrophyte origin. (Panel **B**) shows a complete tabulated list of all ancestral HPPGs (listed by identifier, with the predicted function given in brackets) in which at least one chimerism event between domains of red algal, green algal, aplastidic stramenopile, other

*Figure 6 continued on next page*

*Figure 6 continued*

eukaryotic, and prokaryotic origin was detected. In each case, the inferred evolutionary origins of the N-terminal (NTD) and C-terminal (CTD) components of the chimeric members of the HPPG are given, according to the colour key within the figure, followed by its distribution across all ochrophyte lineages. The two chimeric HPPGs inferred to have arisen in the ochrophyte ancestor are shown in bold text and labelled with horizontal arrows. Exemplar alignments and phylogenies of the two chimeric proteins inferred to have originated in the ochrophyte ancestor are shown in *Figure 6—figure supplements 1–3*.

The following figure supplements are available for figure 6:

**Figure supplement 1.** Alignments of an ochrophyte-specific riboflavin biosynthesis fusion protein.

**Figure supplement 2.** Origins of ochrophyte plastid 3,4-dihydroxy-2-butanone 4- phosphate synthase.

**Figure supplement 3.** An ochrophyte-specific Tic20 fusion protein.

We then considered the origins of the ancestrally dual-targeted ochrophyte proteins. 15 of the 34 putative ancestrally dual-targeted HPPGs were orthologous to HPPGs of clear evolutionary origin; of these, the majority (11/15; 73%) were of red algal, i.e., probable endosymbiont origin (*Figure 7*, panel B; Table S12- sheet 2 [*Dorrell et al., 2016*]). To determine how these dual-targeted HPPGs have altered the biology of the host, we searched for gene families corresponding to aminoacyl-tRNA synthetases within the 7140 non-redundant gene families previously identified to be shared across the ochrophytes (Table S8- sheet 1 [*Dorrell et al., 2016*]). To enable function of the translational machinery, each genome within the ochrophyte cell (i.e., nucleus, mitochondrion, and plastid) requires aminoacyl-tRNA synthetase activity for each amino acid (*Gile et al., 2015*); thus, if any class of aminoacyl-tRNA synthetase is represented by fewer than three genes, then individual tRNA synthetases must support the biology of multiple organelles through dual-targeting. We identified seven classes of tRNA synthetase for which there were only two gene families in the ochrophyte ancestor, one corresponding to a cytosolic enzyme, and the other to an enzyme that was probably dual-targeted to both the mitochondria and plastid. These include five cases in which the dual-targeted tRNA synthetase was of apparent red algal, i.e., endosymbiont origin (*Figure 7*, panel C). Thus, the acquisition of the ochrophyte plastid also altered the biology of the mitochondria, with dual-targeted proteins of endosymbiont origin functionally replacing endogenous mitochondrial-targeted homologues.

## Complex evolutionary origins of CASH lineage plastids

### A pelagophyte/dictyochophyte origin of the haptophyte plastid proteome

We considered whether our dataset provides evidence for any of the other CASH lineage plastids (cryptomonads, haptophytes, or photosynthetic alveolates) originating within the ochrophytes (*Dorrell and Smith, 2011*; *Stiller et al., 2014*; *Ševčíková et al., 2015*), or evidence for gene transfer from ochrophytes into lineages with complex plastids of green algal origin (chlorarachniophytes and euglenids) (*Maruyama et al., 2011*; *Archibald et al., 2003*). In a majority (243/437) of trees in which they could be assigned a clear origin, plastid-targeted proteins from haptophytes resolved at a position within the ochrophyte clade (Materials and methods; *Figure 8*, panel A; Table S4- sheet 5 [*Dorrell et al., 2016*]). All other groups (except for dinotoms, which have well-defined plastids of diatom origin [*Dorrell and Howe, 2015*; *Imanian et al., 2010*]) generally branched externally rather than within the ochrophyte clade (*Figure 8*, panel A). Indeed, the proportion of haptophyte proteins that resolved within the ochrophytes was found to be significantly greater than any of the other groups except for dinotoms (chi-squared, $p < 1 \times 10^{-05}$; Table S4- sheet 5 [*Dorrell et al., 2016*]).

We noted that the plastid-targeted haptophyte proteins of ochrophyte origin were biased towards specific origins, with over half of the proteins that grouped with a specific ochrophyte lineage (100/178) resolving with members of the hypogyristea (i.e., pelagophytes, dictyochophytes, and bolidophytes; *Figure 8—figure supplement 1*; Table S4- sheet 5 [*Dorrell et al., 2016*]). No such bias was observed in any other CASH lineage, in which invariably a significantly smaller proportion of proteins were found to resolve with hypogyristean lineages (chi-squared $p < 0.01$; *Figure 8—figure supplement 1*; Table S4- sheet 5 [*Dorrell et al., 2016*]). We additionally explored whether there

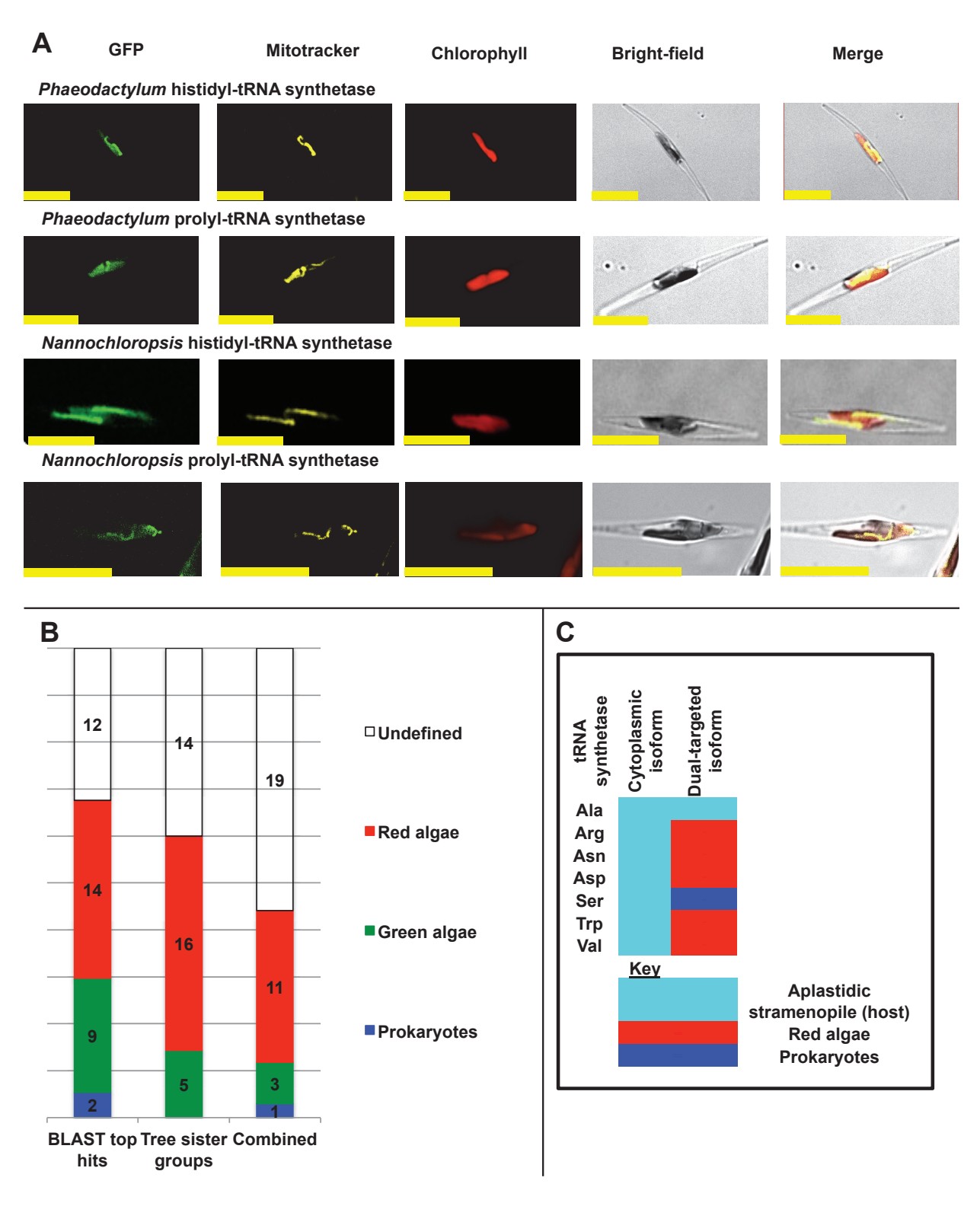

**Figure 7.** Ancient and bidirectional connections between the ochrophyte plastid and mitochondria. (Panel **A**) shows Mitotracker-Orange stained *P. tricornutum* lines expressing GFP fusion constructs for the N-terminal regions of histidyl- and prolyl-tRNA synthetase sequences from *P. tricornutum* and the eustigmatophyte *Nannochloropsis gaditana*. Targeting constructs for an additional four dual-targeted proteins in *P. tricornutum* and one dual-targeted protein in *G. foliaceum*, alongside Mitotracker-negative and wild type control images, are shown in ***Figure 7—figure supplement 1***. (Panel **B**)
*Figure 7 continued on next page*

*Figure 7 continued*

profiles the predicted evolutionary origins of the 34 ancestral dual-targeted HPPGs, as inferred by BLAST top hit and single-gene tree analysis. Data supporting the thresholds used to identify probable dual-targeted HPPGs in silico are supplied in *Figure 7—figure supplement 2*. (Panel C) shows seven classes of tRNA synthetase for which only two copies were inferred in the genome of the last common ochrophyte ancestor. Evolutionary origins are inferred from combined BLAST top hit and single-gene tree analysis for dual-targeted proteins, and from BLAST top hit analysis alone for cytoplasmic proteins. In five cases the dual-targeted isoform is inferred to be of ultimate red algal origin, indicating that a protein derived from the endosymbiont has functionally replaced the endogenous host mitochondria-targeted copy.

The following figure supplements are available for figure 7:

**Figure supplement 1.** Experimental verification of additional ochrophyte dual-targeted proteins.

**Figure supplement 2.** Comparison of different in silico targeting prediction programmes for the identification of dual-targeted ochrophyte proteins.

---

might be unique synapomorphies shared between one ochrophyte lineage and the haptophytes. We found 53 ASAFind-generated HPPGs that contained a majority ($\geq$2/3) of the haptophyte sub-categories and contained at least one member of the hypogyristea, but contained no other ochrophyte orthologues (*Figure 8*, panel B; Table S2- sheet 2, section 3 [*Dorrell et al., 2016*]). This was significantly more than would be expected (28.3, chi-squared p=0.00013) through a random assortment of all HPPGs that were uniquely shared between haptophytes and one ochrophyte lineage, corrected for the relative size of each dataset (Materials and methods). We similarly found a significantly larger number of HPPGs to be uniquely shared between a majority of both the haptophytes and a majority ($\geq$2/3) of the hypogyristean sub-categories (15, expected number 8.0, p=0.034; *Figure 8*, panel B) or shared between a majority of hypogyristea and at least one haptophyte sub-category (28, expected number 12.9, p=0.00073; Table S2- sheet 2, section 3 [*Dorrell et al., 2016*]; *Figure 8*, panel B). Thus, our data supports a specific gene transfer event between the hypogyristea and the haptophytes.

We investigated whether there is a more specific origin for the ochrophyte sequences in haptophyte plastids. First, we tabulated the individual ochrophyte sub-categories identified in the first sister group to haptophyte sequences, of which the greatest number (94) resolved specifically with pelagophyte and dictyochophyte sequences, rather than with bolidophytes, non-hypogyristean lineages, or more ancestral nodes (*Figure 8*, panel C; *Figure 8—figure supplement 2*). Next, we extracted all of the haptophyte plastid-targeted sequences assembled into each ancestral ochrophyte HPPG, performed BLAST top hit analysis (Table S13- sheets 1–3 [*Dorrell et al., 2016*]), and identified sequences for which the best hit was from the same ochrophyte lineage (diatoms, hypogyristea, or chrysista) as the tree sister group (Table S13- sheet 4 [*Dorrell et al., 2016*]). We performed separate analyses for query sequences from each of the three haptophyte sub-categories considered in our analysis (pavlovophytes, prymnesiales, or isochrysidales). In each case, at least 50% of the sequences that produced an evolutionarily consistent series of top hits resolved either with the pelagophytes or dictyochophytes (*Figure 8—figure supplement 3*; Table S13- sheet 4 [*Dorrell et al., 2016*]). Thus, these proteins originated within an ancestor of the pelagophyte/dictyochophyte lineage.

We next tested the probable direction of the gene transfer events. We reasoned that if the genes identified within our study had been transferred from an ancestor of pelagophytes and dictyochophytes into the haptophytes, then we should also see a strong secondary signal linking the haptophytes to earlier ancestors of the pelagophyte/dictyochophyte clade, for example the common ancestor of hypogyristea and diatoms. We inspected the secondary BLAST top hits associated with genes shared between haptophytes and hypogyristea (*Figure 8—figure supplement 4*; Table S13- sheet 5 [*Dorrell et al., 2016*]), and the next deepest sister-groups to haptophyte proteins that are of probable pelagophyte or dictyochophyte origin in each single-gene tree (*Figure 8—figure supplement 4*; Table S4- sheet 2, section 6 [*Dorrell et al., 2016*]). The majority of haptophyte proteins of hypogyristean origin in single-gene trees (65/100) clearly resolved within a broader HPPG containing multiple ochrophyte lineages, and this bias was corroborated by the specific sister groups associated with each protein as inferred by heat map analysis (*Figure 8—figure supplement 4*, panel A). Moreover, the majority of haptophyte proteins with hypogyristean BLAST top hits, and hypogyristean

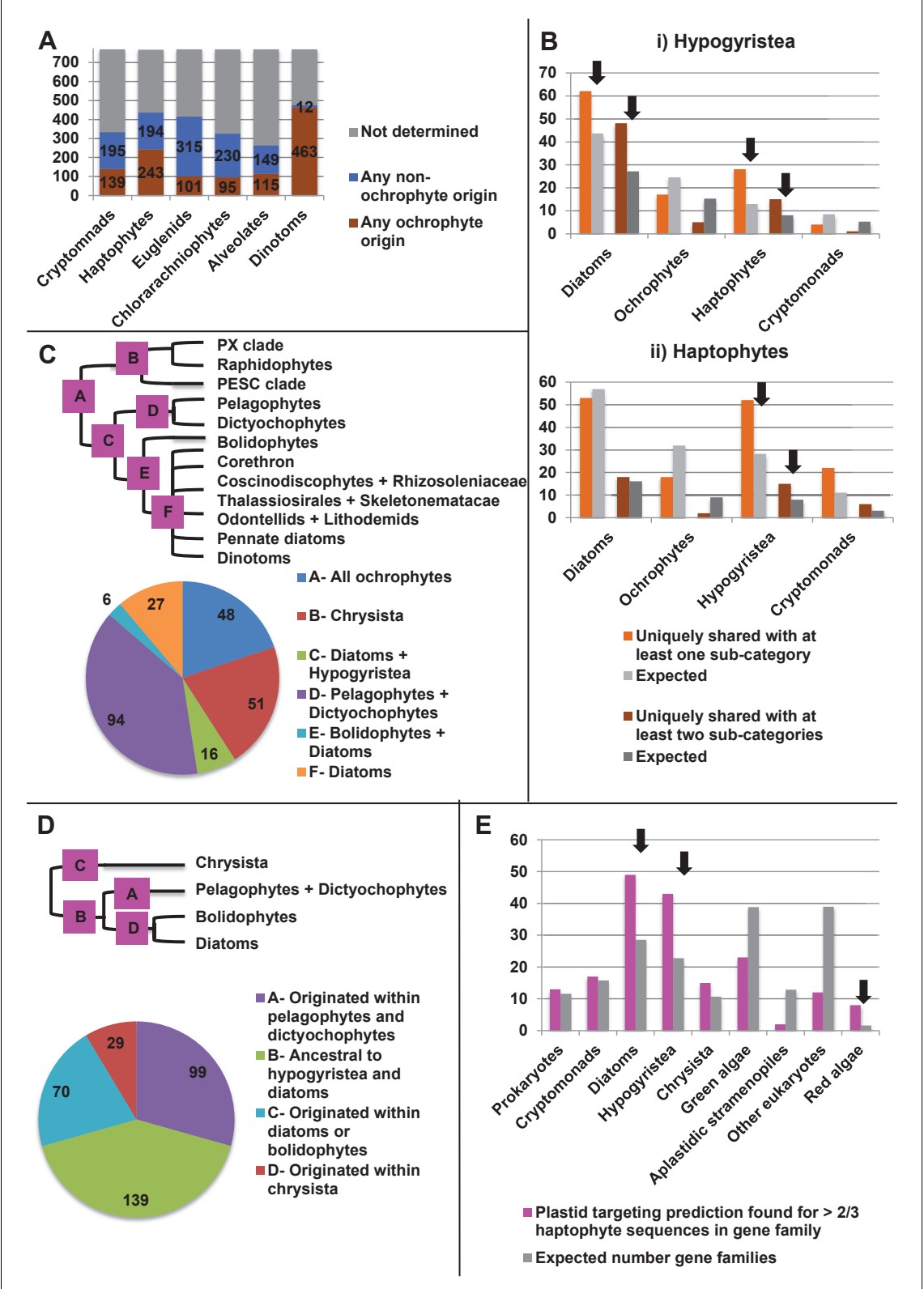

**Figure 8.** Footprints of an ancient endosymbiosis in the haptophyte plastid proteome. (Panel **A**) indicates the number of ancestral ochrophyte HPPGs that included sequences from other algal lineages in single-gene tree analyses, and whether those algal lineages branched within or external to ochrophytes. An overview of the specific origins of proteins of ochrophyte origin in each lineage is shown in *Figure 8—figure supplement 1*. (Panel **B**) compares the number of ASAFind-derived HPPGs that are uniquely shared between hypogyristea (i) or haptophytes (ii) and one other CASH lineage. *Figure 8 continued on next page*

*Figure 8 continued*

Values are given for proteins found in a majority of sub-categories in hypogyristea/ haptophytes and at least one sub-category from only one other lineage (light bars), and proteins found in a majority of sub-categories in hypogyristea/ haptophytes and a majority of sub-categories from only one other lineage (dark bars). Values that are significantly greater than would be expected through random distribution are labelled with black arrows. (Panel **C**) shows a schematic ochrophyte tree, with six different ancestral nodes within this tree labelled with coloured boxes, and the most probable origin point for each of the 243 haptophyte plastid-targeted proteins of probable ochrophyte origin within this tree, as inferred by inspection of the nearest ochrophyte sister-group in single-gene trees. A detailed heatmap of the ochrophyte sub-categories contained in each lineage is shown in *Figure 8—figure supplement 2*, and BLAST top hit analyses corresponding to each plastid-targeted protein are shown in *Figure 8—figure supplement 3*. (Panel **D**) shows the number of residues that are uniquely shared between haptophytes and each node of the ochrophyte tree for 37 genes in which there has been a clear transfer from ochrophytes to haptophytes, and entirely vertical subsequent inheritance. A similar graph, showing the earliest possible inferred origin of each uniquely shared residue, is shown in *Figure 8—figure supplement 4*. (Panel **E**) shows the number of the 12728 conserved gene families inferred to have been present in the last common haptophyte ancestor that are predicted by ASAFind to encode proteins targeted to the plastid, subdivided by probable evolutionary origin, and the number expected to be present in each category assuming a random distribution of plastid-targeted proteins across the entire dataset, independent of evolutionary origin. Evolutionary categories of proteins found to be significantly more likely (chi-squared test, p=0.05) to encode plastid-targeted proteins than would be expected by random distribution are labelled with black arrows. The evolutionary origins of the ancestral gene families are shown in *Figure 8—figure supplement 5*.

The following figure supplements are available for figure 8:

**Figure supplement 1.** Origin of proteins of ochrophyte origin in different CASH lineages.

**Figure supplement 2.** Heatmaps of nearest sister-groups to haptophytes in ancestral ochrophyte HPPG trees.

**Figure supplement 3.** Internal evolutionary affinities of haptophyte plastid-targeted proteins incorporated into ancestral ochrophyte HPPGs.

**Figure supplement 4.** Evidence for gene transfer from pelagophytes and dictyochophytes into haptophytes.

**Figure supplement 5.** Earliest possible origin points of uniquely conserved sites in haptophyte plastid-targeted proteins.

**Figure supplement 6.** Evolutionary origin of ancestral haptophyte genes.

proteins with haptophyte BLAST top hits (48/86 sequences total) had next best BLAST hits against diatoms (*Figure 8—figure supplement 4*, panel B). We additionally tabulated the earliest and latest possible origin points of amino acid residues that were uniquely shared between haptophytes and some but not all ochrophyte lineages, from a dataset of 37 HPPGs for which there was a clear evolutionary affinity between haptophytes and ochrophytes and strict subsequent vertical inheritance (*Figure 8*, panel D; *Figure 8—figure supplement 5*; Table S6- sheets 3, 4 [*Dorrell et al., 2016*]). A greater number of the uniquely shared residues were conserved between the haptophytes and the common ancestor of hypogyristea and diatoms, than were specifically only shared with pelagophyte and dictyochophyte sequences, both per the latest possible origin (139 residues shared with hypogyristea and diatoms; 99 residues with pelagophytes and dictyochophytes; *Figure 8*, panel D; Table S7- sheets 2, 3 [*Dorrell et al., 2016*]) and per the earliest possible origin (46 residues shared with hypogyristea and diatoms; 41 residues with pelagophytes and dictyochophytes; *Figure 8—figure supplement 5*; Table S7- sheets 2, 3 [*Dorrell et al., 2016*]). This specifically supports a transfer of plastid-targeted proteins from an ancestor of the pelagophyte/dictyochophyte clade into the haptophytes, rather than the other way around.

Finally, we tested whether these proteins were likely to have been acquired through an endosymbiotic event. We reasoned that the genes acquired by haptophytes through endosymbiotic events should encode a greater proportion of plastid-targeted proteins than would be observed with genes of alternative origin. We accordingly constructed a dataset of 12,728 non-redundant gene families that were broadly distributed across the haptophytes (Table S14- sheet 1 [*Dorrell et al., 2016*]), of which 772 were of probable hypogyristean origin (*Figure 8—figure supplement 6*; Table S14- sheet 2 [*Dorrell et al., 2016*]). A significantly larger proportion of the ancestral haptophyte gene families of hypogyristean origin were predicted by ASAFind to be targeted to the plastid than would be expected by random distribution of the data (observed number 43, expected number 22.8, chi-squared p=$2.2 \times 10^{-05}$; *Figure 8*, panel E; Table S14- sheet 3 [*Dorrell et al., 2016*]), consistent with

an endosymbiotic origin. Thus, our data support an endosymbiotic uptake of an ancestor of the pelagophytes and dictyochophytes by an ancestor of the haptophytes.

## Phylogenetic discrepancies between the haptophyte plastid proteome and genome

The transfer of plastid-targeted proteins from the pelagophyte/dictyochophyte clade into the haptophytes is surprising, as previous studies have indicated that the haptophyte plastid genome originates either as a sister-group to the entire ochrophyte lineage (*Stiller et al., 2014*) or to the cryptomonads (*Khan et al., 2007*; *Le Corguillé et al., 2009*; *Muñoz-Gómez et al., 2017* . To verify this discrepancy we constructed two plastid trees, one using 54 conserved proteins that are encoded in all sequenced red lineage and glaucophyte plastids (*Figure 9*, panel A; Table S15- sheet 1 [*Dorrell et al., 2016*]), and one using a smaller subset of 10 plastid-encoded proteins that were detected in many of the transcriptome libraries used in this study (*Figure 9*, panel B; Table S15- sheet 1 [*Dorrell et al., 2016*]).

A specific sister-group relationship between the cryptomonads and haptophytes was recovered, with moderate to strong bootstrap support, in both the gene-rich tree (*Figure 9*, panel A) and the taxon-rich tree (*Figure 9*, panel B). Both trees also strongly supported the monophyly of ochrophyte plastid genomes (*Figure 9*). Alternative topology tests rejected any possibility that the haptophyte plastid originated within the ochrophytes (*Figure 9—figure supplement 1*; p≤0.05). Similarly, trees calculated from alignments in which fast-evolving sites and clades had been serially removed, and in which the alignment had been recoded to minimise amino acid composition biases (*Figure 9—figure supplement 2*; Table S15- sheet 2; Table S16 [*Dorrell et al., 2016*]) either recovered a sister-group relationship between haptophytes and cryptomonads, or placed haptophytes as the sister group to all ochrophytes. We additionally generated and inspected single-gene tree topologies for each of the constituent genes used to generate each concatenated multigene alignment, and could not find any that confidently resolved a sister-group relationship between haptophytes and the pelagophyte/dictyochophyte clade (*Figure 9—figure supplement 3*; Table S15- sheet 3 [*Dorrell et al., 2016*]). Finally, we found only three residues in the alignment that were uniquely shared among all four haptophytes and the sole representative of pelagophytes and dictyochophytes (*Aureococcus*) in the gene-rich dataset, and no residues that were shared between a majority of the haptophytes and at least one pelagophyte or dictyochophyte sequence in the taxon-rich dataset (*Figure 8*, panel C; Table S17- sheet 4 [*Dorrell et al., 2016*]). In contrast, we found large numbers of residues that were shared uniquely by haptophytes and other lineages (*Figure 9*, panel C; Table S17- sheet 4 [*Dorrell et al., 2016*]). This strong support for a relationship between haptophytes and cryptomonads is inconsistent with phylogenetic artifacts such as coevolution between specific protein complexes (*Dorrell et al., 2017*; *Guo and Stiller, 2005*) or gene duplication and differential loss of paralogues (*Qiu et al., 2012*), in which case there should still be a detectable underlying signal linking it to the pelagophytes and dictyochophytes. We conclude that while many plastid-targeted haptophyte proteins originate from an ancestor of the pelagophytes and dictyochophytes, the haptophyte plastid genome does not.

## Discussion

In this study, we have reconstructed an experimentally verified dataset of 770 plastid-targeted proteins that were present in the last common ancestor of all ochrophytes (*Figures 1* and *2*). Our dataset accordingly provides windows into the evolutionary origins of the ochrophyte plastid lineage. These include evidence for a green algal contribution to ochrophyte plastid evolution and a late acquisition of the ochrophyte plastid following divergence of the ochrophyte lineage from oomycetes (*Figures 3* and *4*). Although each of these findings have been previously suggested by studies of whole stramenopile genomes (*Moustafa et al., 2009*; *Stiller et al., 2009*) our data represent to our knowledge the first large-scale verification from studies of plastid targeted proteins for both of these important events in the origins of the ochrophyte plastid.The relatively late origin of the ochrophyte plastid is particularly interesting as molecular divergence estimates place the ochrophytes as diverging from the oomycetes no more than 90 million years prior to the radiation of ochrophyte lineages (*Brown and Sorhannus, 2010*; *Matari and Blair, 2014*). Assuming that these estimates are

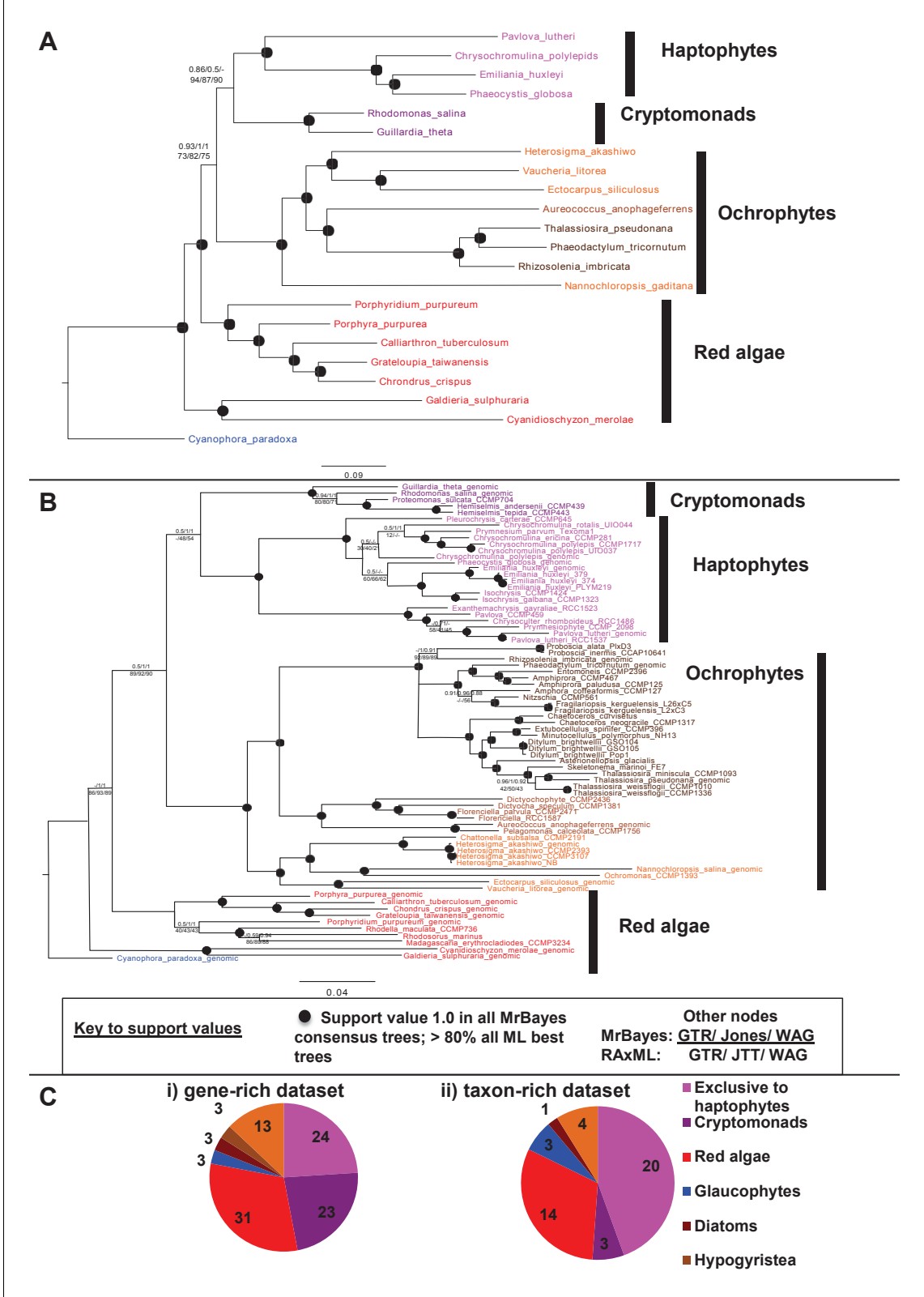

**Figure 9.** Non-ochrophyte origins of the haptophyte plastid genome. (Panels **A** and **B**), respectively, show gene-rich and taxon-rich phylogenies of plastid-encoded proteins from red algae and plastids of red algal origin with the glaucophyte *Cyanophora paradoxa* as outgroup. (Panel **A**) Combined Bayesian and Maximum Likelihood analysis (MrBayes + RAxML, GTR, JTT, WAG) of a 22 taxa x 12103 aa alignment of 54 proteins encoded by all published red and red-derived plastid genomes. (Panel **B**) analysis of a 75 taxa x 3737 aa alignment of 10 conserved plastid-encoded proteins

*Figure 9 continued on next page*

*Figure 9 continued*

detectable in a broad range of red lineage MMETSP libraries. Nodes resolve with robust support (posterior probabilities of 1 for all Bayesian trees and >80% bootstrap support for all ML trees) are shown with filled circles; individual support values for each analysis are shown for the remaining nodes are shown as detailed in the box below panel B. Alternative topology tests, the results of fast-site and clade deduction analysis for each tree, and heatmap comparisons of sister-group relationships identified for single-gene trees of each constituent gene within each concatenated alignment are shown in *Figure 9—figure supplements 1–3*. (Panel C) shows the number of residues in each alignment that are uniquely shared between haptophytes and only one other lineage. For the gene-rich alignment (i), which is gap-free, residues are included that are found in all four haptophyte sequences and at least one sequence from the lineage under consideration. For the taxon-rich alignment (ii), to account for the presence of gapped positions, residues are included that are found in at least 11 of the 22 haptophyte sequences and at least one sequence from the lineage under consideration.

The following figure supplements are available for figure 9:

**Figure supplement 1.** Alternative topology tests of plastid genome trees.

**Figure supplement 2.** Fast site removal and clade deduction analysis of plastid genome trees.

**Figure supplement 3.** Single-gene tree topologies associated with individual plastid-encoded genes.

reliable, our dataset represents some of the earliest proteins to support the ochrophyte plastid following its endosymbiotic uptake. We also provide evidence for widespread mixing of proteins of different evolutionary origin in the ancestral ochrophyte plastid (*Figure 5*), including evidence for the formation of new fusion proteins through the recombination of domains of different evolutionary origins (*Figure 6*), and a bidirectional interaction between proteins derived from the endosymbiont with proteins from host organelles via dual-targeting (*Figure 7*). A schematic outline of these results is shown in *Figure 10*.

Many questions nonetheless remain to be answered. It remains to be determined whether the in silico prediction facilitated by programmes such as ASAFind and HECTAR are sufficient to enable the identification of all ochrophyte plastid proteins (*Gruber et al., 2015*; *Gschloessl et al., 2008*). This is particularly pertinent in the context of dual-targeted proteins, insofar as the dataset of 34 potentially ancestrally dual-targeted proteins identified in this study may not include proteins that are dual-targeted to the plastid and other cellular organelles, such as the ER (*Porter et al., 2015*), cytoplasm (*Pham et al., 2014*), or nucleus (*Krause et al., 2012*). We note also that, based on the fluorescence patterns observed with the exemplar proteins within this study (*Figures 2* and *7*), ASA-Find and HECTAR may identify proteins targeted to the periplastid compartment, as well as to the plastid stroma. While these periplastid and multipartite proteins probably form an important part of plastid physiology, it will be interesting to dissect the specific signals associated with the targeting of proteins to individual sub-compartments within CASH lineage plastids (*Tanaka et al., 2015a*; *Liu et al., 2016*).

Another major question concerns the origins of plastid-targeted proteins of green algal origin in ochrophytes. Overall, our data supports the targeting of a significant complement of proteins of chlorophyte origin to the ochrophyte plastid (*Figure 4*). It remains to be determined, however, what the exact chlorophyte donor was, and how these genes may have been acquired. It is possible that the green genes were transferred into the ochrophyte lineage via lateral gene transfer, either from a range of different green algal sources or repeatedly from one lineage (for example, a semi-permanent intracellular symbiont [*Dorrell and Howe, 2012a*]), although neither scenario would explain the bias in green algal genes in ochrophyte genomes towards encoding proteins of plastid function (*Figure 4*, panel D). An alternative possibility might be a cryptic green algal endosymbiosis in the evolutionary history of the host, as has been previously suggested (*Dorrell and Smith, 2011*; *Moustafa et al., 2009*) (*Figure 10*), or a more convoluted pattern of acquisition. We note, for example, that the green genes identified in our study are not only plastid-targeted across the ochrophytes, but are apparently shared with haptophytes and cryptomonads (*Figure 10—figure supplement 1*), which would be equally consistent with them having been present in a common ancestor of the CASH lineage plastid, and relocated to each host nuclear lineage following endosymbiosis (*Figure 10*). Thus, pinpointing the exact nature and timing of the green gene transfer into ochrophytes rests not only on more extensive sequencing of deep-branching chlorophyte lineages,

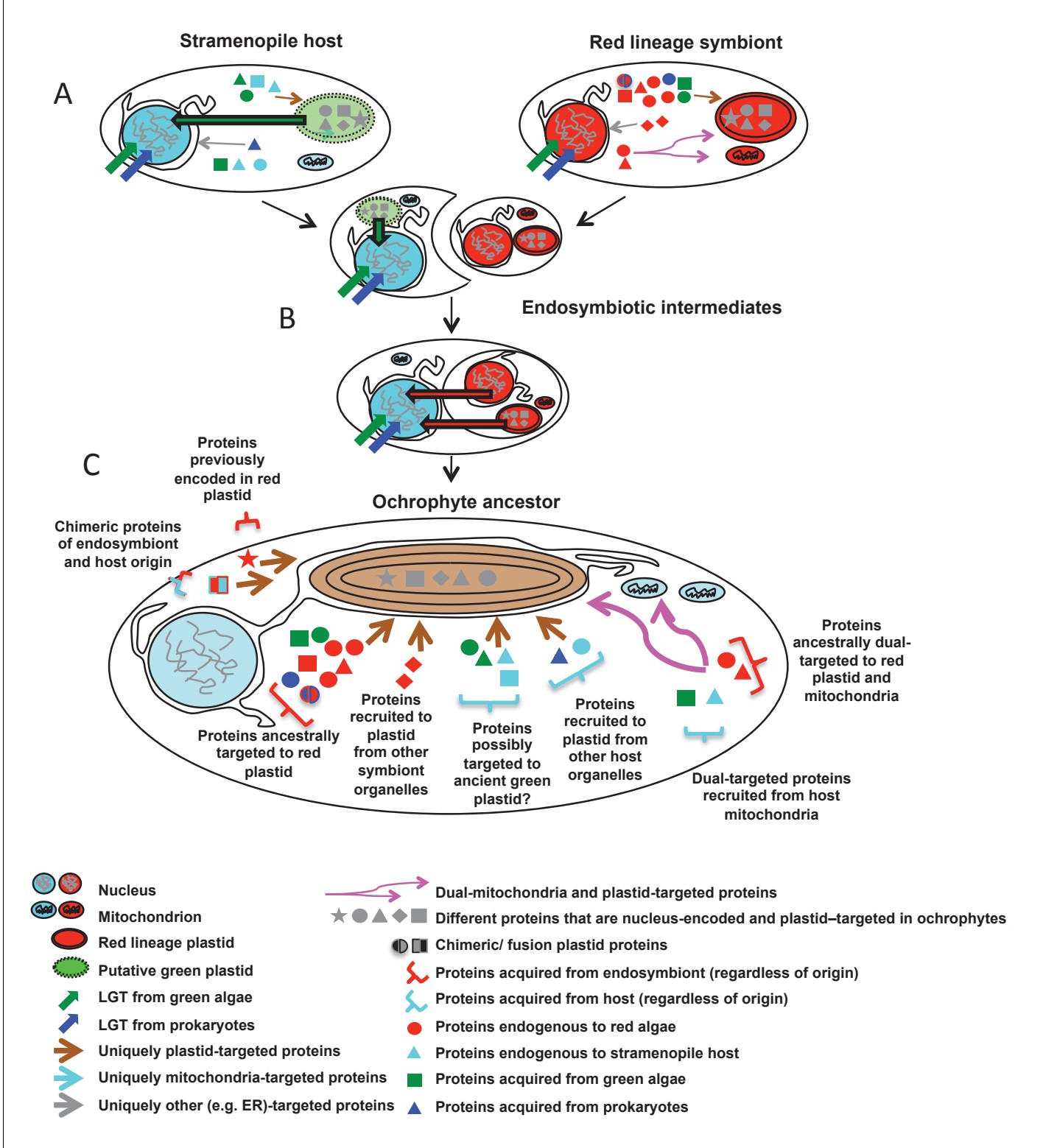

**Figure 10.** Schematic diagram of events giving rise to the ancestral ochrophyte plastid proteome. Each cell diagram depicts a different stage in the ochrophyte plastid endosymbiosis; each protein depicted represents one or more proteins inferred in this study to have been nucleus-encoded and plastid-targeted in the last common ancestor of all ochrophytes. An ancient ochrophyte ancestor, which had already diverged from oomycetes and other aplastidic stramenopile relatives, and which may have possessed a green algal plastid (A), acquired a red lineage plastid via secondary or higher endosymbiosis (B). Both the host and the endosymbiont are likely to have been evolutionary chimeras, possessing proteins encoded by genes acquired

*Figure 10 continued on next page*

*Figure 10 continued*

from endosymbiotic and/or lateral gene transfer events. Both host and symbiont are additionally likely to have possessed chimeric proteins, generated through the fusion of genes of different evolutionary origins, and a large number of mitochondrial-, ER- and (in the case of the red endosymbiont) potentially dual-targeted proteins. Following genetic integration of the red endosymbiont with its stramenopile host, the first ochrophytes (C) thus possessed a wide range of proteins of plastid function acquired from different sources, with no apparent functional bias in the types of proteins that were retained from different sources. Chimeric proteins and dual-targeted proteins, either acquired directly from the endosymbiont, or generated *de novo*, were also widespread features of this ancestral plastid proteome. Detailed information regarding the relationship between ultimate the evolutionary origins of each HPPG, and its presence or absence in other CASH lineages, is provided in *Figure 10—figure supplement 1*. A schematic diagram of possible models through which the haptophyte plastid may have originated is shown in *Figure 10—figure supplement 2*.

The following figure supplements are available for figure 10:

**Figure supplement 1.** Complex origins of different ancestral ochrophyte HPPGs.

**Figure supplement 2.** Different scenarios for the origins of haptophyte plastids.

but also on characterising the genome composition of the closest aplastidic relatives of extant ochrophytes (e.g., *Develorapax, Pirsonia* [*Aleoshin et al., 2016*]), and the closest red algal relative of CASH lineage plastids, which remains unknown (*Dorrell and Smith, 2011*; *Baurain et al., 2010*).

We also provide evidence for a chimeric origin of the haptophyte plastid (*Figures 8* and *9*). A schematic outline of these results is shown in *Figure 10—figure supplement 2*. We have shown that a significant number of plastid-targeted proteins found in haptophytes originate from an ancestor of the pelagophytes and dictyochophytes (*Figure 8*). Although it has previously been suggested from studies of nuclear genomes that ochrophyte and haptophyte plastids share a close evolutionary history, (*Stiller et al., 2014*; *Miller and Delwiche, 2015*) it has not previously been shown robustly that the haptophyte plastid resolves at a specific position internal to the ochrophyte lineage. This data supports findings from other studies (such as a possible origin for the plastids of dinoflagellates and apicomplexans within a member of the PESC clade; *Ševčíková et al., 2015*) that many of the plastids found within CASH lineages are of tertiary or higher endosymbiotic origin. The at least partial pelagophyte/dictyochophyte origin of the haptophyte plastid is supported by multiple lines of evidence- i.e., uniquely shared proteins, single-gene tree topologies, BLAST top hit analysis, and analysis of synapomorphies in multigene alignments (*Figure 8* and supplements). Alongside the bias of haptophyte genes of hypogyristean origin encoding proteins of plastid function (*Figure 8*- panel E), these observations argue against these genes having been acquired through multiple independent lateral gene transfer events, and instead support an endosymbiosis event. We note that other studies have shown strong evidence for gene transfers between haptophytes and individual members of the hypogyristea: for example, Stiller *et al.* have demonstrated a strong enrichment in BLAST top hits against haptophytes, from the genome of the pelagophyte *Aureococcus anophagefferens,* compared to other ochrophyte genomes (*Stiller et al., 2014*). We additionally note that an ancestral gene transfer from a pelagophyte/dictyochophyte ancestor into the haptophytes is a chronologically realistic scenario: molecular clock estimates place the pelagophytes and dictyochophytes diverging between 300 and 700 million years before present (*Brown and Sorhannus, 2010*; *Parfrey et al., 2011*), which broadly overlaps with the molecular dates estimated for the radiation of the haptophytes in the same studies (*Brown and Sorhannus, 2010*; *Parfrey et al., 2011*), and precedes the first haptophyte microfossils, identified ca. 220 million years before the present (*Bown, 1998*).

Finally, we verify that the evolutionary links between haptophyte and the pelagophyte/dictyochophyte clade in terms of plastid-targeted proteins are not supported by phylogenies of the haptophyte plastid genome (*Figure 9*). Other multigene phylogenies of red lineage plastid genomes have similarly demonstrated that the haptophyte plastid genome instead resolves as a sister-lineage either to cryptomonads or to all ochrophytes (*Stiller et al., 2014*; *Janouskovec et al., 2010*; *Khan et al., 2007*; *Le Corguillé et al., 2009*). Furthermore, the structure and content of haptophyte and hypogyristean plastid genomes are dissimilar: for example, haptophyte plastids possess an *rpl36* gene that has been laterally acquired from a bacterial donor and is shared with cryptomonad plastids but absent from ochrophytes (*Rice and Palmer, 2006*), and ochrophyte plastids no longer retain genes encoding the plastid division machinery proteins *minD* and *minE*, which remain plastid-

encoded in haptophytes and cryptomonads (*de Vries and Gould, 2017*). Finally, extant haptophyte plastids have comparatively large plastid genomes and possess a conventional quadripartite structure (*Green, 2011*), whereas sequenced pelagophyte plastids (the harmful coastal species *Aureococcus anophagefferens* and *Aureoumbra lagunensis*, and an uncultured member of the predominantly open ocean genus *Pelagomonas*) all have a reduced coding content compared to other photosynthetic ochrophytes, cryptomonads and haptophytes, and have secondarily lost the plastid inverted repeat (*Worden et al., 2012*; *Ong et al., 2010*), although it is not yet known whether the plastid genomes of other pelagophyte genera and of dictyochophytes share this reduced structure.

The discrepancy between the pelagophyte/dictyochophyte origin of the haptophyte plastid proteome and the clear non-ochrophyte origin of its plastid genome might be explained by several different evolutionary scenarios. One possibility would be a serial endosymbiosis event deep in haptophyte evolutionary history, in which an ancient plastid derived from a pelagophyte/dictyochophyte ancestor was acquired by the haptophyte common ancestor, then replaced subsequently by a plastid of non-ochrophyte origin (*Figure 10—figure supplement 2*). This discrepancy, alongside others such as the presence of green algal genes in ochrophytes, bolsters the possibility that serial plastid endosymbiosis has been a widespread component of the evolution of CASH lineage plastids other than the dinoflagellates, in which it is a well established phenomenon (*Dorrell and Howe, 2015*; *Yamada et al., 2017*). Verifying this scenario, or its alternatives (such as lateral gene transfer from pelagophyte or dictyochophyte algae into the algal ancestors of the haptophyte plastid) rests on identifying the exact origin of the current haptophyte plastid genome, and in particular demonstrating that the haptophyte plastid genome originates from within (rather than forms a sister-group to) a major lineage of eukaryotic algae other than ochrophytes (*Figure 10—figure supplement 2*). For this, sequence data from early-diverging members of the cryptomonads and haptophytes will be particularly important (*Yabuki et al., 2014*; *Choi et al., 2017*; *Kim et al., 2011*). It also remains to be determined whether other CASH lineage plastids, such as the peridinin-type plastids found in most photosynthetic alveolates, originate within the ochrophytes (*Ševčíková et al., 2015*; *Dorrell and Howe, 2015*). Similar plastid proteome reconstructions, using bespoke datasets for these species, will be particularly useful in unravelling their disparate evolutionary origins.

Overall, our dataset provides valuable and deep insights into the chimeric origins and complex fates of a major group of eukaryotic algae. Further studies using more sensitive pipelines, or using analogous datasets from other major CASH lineages, may elucidate the evolutionary and physiological diversification of plastids across the eukaryote tree of life.

## Materials and methods

### Identification of ancestral plastid-targeted ochrophyte proteins

Ancestral plastid-targeted proteins in ochrophytes were identified via a composite pathway, consisting of in silico prediction, identification of conserved proteins using BLAST, alignment, and single-gene tree building. First, the complete protein libraries annotated from eleven ochrophyte genomes (the diatoms *Phaeodactylum tricornutum* (*Bowler et al., 2008*), *Thalassiosira pseudonana* (*Armbrust et al., 2004*), *Thalassiosira oceanica* (*Lommer et al., 2012*), *Fistulifera solaris* (*Tanaka et al., 2015b*), *Fragilariopsis cylindrus*, *Synedra acus* (*Galachyants et al., 2015*), and *Pseudonitzschia multiseries*; the pelagophyte *Aureococcus anophage*fferens (*Gobler et al., 2011*); the eustigmatophytes *Nannochloropsis gaditana* and *Nannochloropsis salina* (*Radakovits et al., 2013*; *Wang et al., 2014*); and the kelp *Ectocarpus siliculosus* (*Cock et al., 2010*); Table S1- sheet 1 [*Dorrell et al., 2016*]), were screened using the ochrophyte plastid-targeting predictors ASAFind (*Gruber et al., 2015*) (used in conjunction with SignalP version 3.0 (*Bendtsen et al., 2004*); Table S2 [*Dorrell et al., 2016*]) and HECTAR (*Gschloessl et al., 2008*) (integrated into a Galaxy (*Afgan et al., 2016*) instance available at http://webtools.sb-roscoff.fr; Table S3 [*Dorrell et al., 2016*]). All proteins that were deemed to possess plastid-targeting sequences (regardless of the confidence score applied by ASAFind [*Gruber et al., 2015*]) were retained for further inspection.

Possible conserved plastid-targeted sequences (i.e. homologous plastid-targeted protein groups, or HPPGs) were next identified using a customised BLAST protocol. First, a library of non-redundant proteins was generated to serve as seed sequences for further searches. Each plastid-targeted protein identified from ochrophyte genome sequences was searched by BLASTp against a modified

Uniref (*Suzek et al., 2007*) library, and the expect values for all top hits were extracted, to yield a floating BLAST threshold below which orthologous proteins were identified. All sequences from lineages with a history of secondary endosymbiosis were first removed from the Uniref library in order to avoid the confounding effects of gene transfer from current and former symbionts (*Stiller et al., 2014*; *Ševčíková et al., 2015*; *Maruyama et al., 2011*; *Archibald et al., 2003*). The removed lineages included cryptomonads, centrohelids, telonemids, haptophytes, alveolates, rhizaria, euglenids, and plastid-bearing stramenopiles. All of the ochrophyte genome-derived plastid-targeted proteins were searched against one another by BLAST, and proteins that matched one another with an expect score lower than the first outgroup hit (or were retrieved as a stronger match than the outgroup hit if the expected values of both were zero), and thus likely correspond to different proteins within the same monophyletic plastid protein cluster, were merged. Only one protein was retained as the seed sequence for subsequent growth of each cluster: this was defined first via organism (in order of preference: *P. tricornutum, T. pseudonana, P. multiseries, F. cylindrus, S. acus, A. anophageferrens, E. siliculosus, N. gaditana, N. salina, T. oceanica, F. solaris*) and, where more than one protein was available for a given organism, the protein with the lowest BLAST expect value against the corresponding uniref top hit.

Next, plastid-targeted protein sequences were sought from all available ochrophyte sequence data. A search database was built from all eleven completed ochrophyte genomes, 147 ochrophyte sequence libraries from the Marine Microeukaryote Transcriptome Sequence Project (MMETSP) (*Keeling et al., 2014*), eleven further ochrophyte transcriptome sequencing projects (*Matasci et al., 2014*; *Mangot et al., 2017*; *Kessenich et al., 2014*) and uniref. Cross-contamination was removed from MMETSP transcriptomes as previously described (*Marron et al., 2016*). Briefly, this procedure compares the nucleotide sequences of contigs assembled from each MMETSP library by pairwise BLAST, and defines a separate cross-contamination threshold for each pair of MMETSP libraries based on their distribution of BLAST percent identities. These distributions should each contain a peak centered on the average nucleotide percent identity of transcripts between the two species. In addition, in the presence of cross-contamination, there should be a second peak at 100% identity. The procedure defines the cross-contamination threshold as the minimum between these two peaks; above the threshold, contigs (and the proteins predicted from them) are considered to be potentially cross-contaminated. In total, 2.5% of the MMETSP contigs were discarded through this method. A summary of the number of contigs discarded is provided in Table S1- sheet 2, section 1 (*Dorrell et al., 2016*).

Each decontaminated sequence was trimmed at the N-terminus to the first methionine present, and binned into one of eleven different evolutionary categories, based on recent multigene phylogenetic trees for ochrophytes and diatoms (*Derelle et al., 2016*; *Sorhannus and Fox, 2012*; *Yang et al., 2012*; *Theriot et al., 2015*) (*Figure 1*, panel A; Table S1- sheet 1 [*Dorrell et al., 2016*]). These consisted of: three chrysistan lineages (the 'PX clade' of phaeophytes, xanthophytes and related lineages; raphidophytes; and the 'PESC clade' of pinguiophytes, eustigmatophytes, synchromophytes, and synurophytes/chrysophytes), three hypogyristean lineages (pelagophytes; dictyochophytes; and bolidophytes), and five diatom lineages (the basally divergent genus *Corethron*; radial centric lineages such as Coscinodiscophytes and Rhizosoleniaceae; the polar centric Thalassiosirales and Skeletonemataceae, which appear to be relatively distantly related to pennate diatoms (*Sorhannus and Fox, 2012*; *Theriot et al., 2015*); polar centric lineages such as Odontellids and Chaetocerotales that appear to be more closely related to pennate diatoms (*Sorhannus and Fox, 2012*; *Theriot et al., 2015*); and finally all pennate lineages). These binned sequences were then searched for plastid-targeted proteins by ASAFind and HECTAR as before.

The seed sequences for the resulting non-redundant HPPGs were searched against the enlarged plastid sequence library using BLASTp. Proteins that matched against seed sequences with a lower expect value than the outgroup best hit (or were retrieved as a stronger match than the outgroup hit if the expected values of both were zero), were added to each HPPG. Next, three custom thresholds were defined that were particularly successful in distinguishing probable proteins of true plastid localisation from false positives (*Figure 1*, panel B). For this, conservation patterns were selected that maximised the relative enrichment in proteins with unambiguous plastid functions (i.e., were annotated to function in photosynthesis, to constitute integral parts of the plastid thylakoid or inner membranes, or corresponded to the expression products of genes that are plastid-encoded in red algae but have been apparently relocated to the ochrophyte nucleus [*Green, 2011*] or that

corresponded to proteins previously verified experimentally to localise to ochrophyte plastids [*Gruber et al., 2015*; *Gschloessl et al., 2008*; *Huesgen et al., 2013*; *Grouneva et al., 2011*]), and thus should contain relatively fewer examples of mispredicted proteins within the dataset. At the same time, conservation patterns were selected that minimised the number of HPPGs identified as conserved from a negative control dataset (consisting of HPPGs assembled using seed sequences from the published genome sequences of the cryptomonad *Guillardia theta* (*Curtis et al., 2012*) or the haptophytes *Emiliania huxleyi* (*Read et al., 2013)* and *Chrysochromulina tobin* (*Hovde et al., 2015*), and for which no plastid-targeted orthologues were detected in any of the ochrophyte genome sequences used in this study). The thresholds corresponded to: orthologues in a majority (≥2/3) of chrysistan and a majority (≥3/5) of diatom lineages; a majority of chrysistan and a majority (≥2/3) of hypogyristean lineages; and at least one chrysistan, and a majority of both hypogyristean and diatom lineages (*Figure 1*).

All of the HPPGs that passed at least one threshold were extracted, and homology for each HPPG was confirmed individually (Table S4- sheet 1 [*Dorrell et al., 2016*]). First, each HPPG was aligned using 20 iterations of MUSCLE v8 (*Edgar, 2004*), followed by the in-built alignment programme integrated into GeneIOUS v 4.76 (*Kearse et al., 2012*), under the default criteria. Each HPPG alignment was manually inspected, and proteins that failed to align with the genomic sequences, clearly terminated within the conserved region of the protein, or were truncated at the N-terminus by a length of greater than 50 amino acids (i.e. the approximate length of an ochrophyte plastid-targeting sequence [*Gruber et al., 2015*; *Huesgen et al., 2013*]) were removed, following which HPPGs that no longer passed the taxonomic criteria defined for conservation were eliminated (Table S4- sheet 1 [*Dorrell et al., 2016*]). Next, each HPPG was enriched with the sequences for the top 50 hits obtained when the seed sequence was searched against the modified uniref library as detailed above, alongside the single best hit for composite transcriptome and genome libraries constructed for 36 eukaryotic sub-categories (Table S1- sheet 1 [*Dorrell et al., 2016*]), and realigned against this reference. The transcriptome components of the reference sequence libraries were cleaned of residual contamination as defined above, and 23 individual MMETSP libraries were additionally excluded due to evidence of further contamination (Table S1- sheet 2 [*Dorrell et al., 2016*]). Sequences that failed to align were removed, and HPPGs that failed to meet the criteria for conservation following alignment were eliminated (Table S4- sheet 1 [*Dorrell et al., 2016*]).

Finally, each HPPG was trimmed at the N- and C-termini to (respectively) the first residue and last residue visually identified to be conserved in >70% of the sequences in the alignment, corresponding to the probable conserved domain of the protein. Each HPPG was then trimmed with trimAl using the -gt 0.5 option (*Capella-Gutiérrez et al., 2009*). 100 trees were calculated for each trimmed alignment using RAxML, with the JTT substitution model + gamma correction (*Stamatakis, 2014*). The consensus tree from the 100 bootstrap replicates was manually inspected for the presence of a clade of ochrophyte proteins, containing sufficient sequences to pass the criteria for conservation defined above, that was either monophyletic, or paraphyletic to the inclusion of only one of five different non-ochrophyte groups (prokaryotes, red algae, green algae, aplastidic stramenopiles, and all other eukaryotes excluding CASH lineages, rhizaria and euglenids; Table S4- sheet 1 [*Dorrell et al., 2016*]). HPPGs that passed this final stage of analysis were deemed to correspond to ancestrally plastid-targeted proteins (Table S4- sheet 2 [*Dorrell et al., 2016*]).

All identified plastid-targeted proteins, HPPGs, full aligned HPPGs, and single-gene trees have been made publically accessible through the University of Cambridge dSpace server (https://www.repository.cam.ac.uk/handle/1810/261421 [*Dorrell et al., 2016*]).

## Generation of fluorescence expression constructs for *Phaeodactylum tricornutum*

*Phaeodactylum tricornutum* 1.86 (CCMP2561), *Nannochloropsis gaditana* CCMP526, and *Glenodinium foliaceum* PCC499 were maintained in liquid cultures of f/2 medium supplemented with vitamins, and 100 μg/ ml each of ampicillin, streptomycin, kanamycin and neomycin, in a constant 19°C environment in a 12 hr: 12 hr cycle of 150 μE m$^{-2}$ s$^{-1}$ light: dark. *P. tricornutum* was maintained on an orbital shaker at 100 rpm, while *N. gaditana* and *G. foliaceum* were maintained as stationary cultures. Large volume cultures of *P. tricornutum* (e.g. cultures grown for transformation by bombardment) were grown in artificial seawater, supplemented with vitamins but without antibiotics.

Total cellular RNA was extracted from c. 30 ml volumes of late log phase culture from each species using a modified Trizol phase extraction and DNase treatment protocol as described elsewhere (*Dorrell and Howe, 2012b*). Each RNA sample was tested for integrity by gel electrophoresis and quantified by a nanodrop spectrophotometer, and confirmed to be free of residual DNA contamination by direct PCR using universal eukaryotic 18S rDNA primers (*Gachon et al., 2013*). Approximately 200 ng purified RNA from each species was used as the template for cDNA synthesis, using a Maxima First Strand cDNA Synthesis Kit (Thermo, France), following the manufacturer's instructions.

Nucleotide sequences encoding plastid-targeted proteins of unusual provenance were identified using the complete genome sequences of *Phaeodactylum tricornutum* and *Nannochloropsis gaditana* (*Radakovits et al., 2013*; *Bowler et al., 2008*), and the *Glenodinium foliaceum* CCAP1116/3 transcriptome library assembled as part of MMETSP (*Keeling et al., 2014*; *Hehenberger et al., 2016*) (Table S5 [*Dorrell et al., 2016*]). Two primers were designed for each sequence: a PCR forward primer corresponding to the 5' end of the ORF, and a translationally in-frame PCR reverse primer positioned a minimum of 45 bp into conserved domain of the protein sequence (Table S5 [*Dorrell et al., 2016*]). These primers were respectively fused to 5' fragments complementing the 3' end of the *P. tricornutum* FcpA promoter, and the 5' end of the GFP CDS. For one gene (the novel plastid protein), PCR reverse primers were designed complementary to the 3' end of the CDS of each gene due to the lack of a verifiable CDD; a full-length PCR reverse primer was additionally designed against the histidyl-tRNA synthetase sequence from *Nannochloropsis gaditana* due to failure to obtain functional expression from N-terminal constructs (data not shown).

High-fidelity PCR products were amplified with each primer pair from the corresponding cDNA product using Pfu DNA polymerase (Thermo, France), per the manufacturer's instructions. In two cases (*Nannochloropsis gaditana* peroxisomal membrane protein, and the novel plastid protein) inserts were amplified from synthetic, codon-optimised constructs, designed to maximise expression levels in *Phaeodactylum tricornutum* (Eurofins, France). Each product was separated by DNA gel electrophoresis, cut, purified using a PCR gel extraction column kit (Macherey-Nagel, France), quantified using a nanodrop spectrophotometer, and verified by Sanger sequencing (GATC Biotech, France). The purified products were then used for Gibson ligation reactions (*Gibson et al., 2009*) (NEB, France), following the manufacturer's instructions, using linearised and DpnI-treated vector sequence generated from the pPhat-eGFP vector (*Siaut et al., 2007*), and transformed into chemically competent Top10 *E. coli* cells, prior to selection on LB-1% agar plates containing 100 µg/ ml ampicillin. Individual colonies were picked, verified to contain the insert sequence by PCR, and grown as overnight liquid cultures on LB medium supplemented with 100 µg/ ml ampicillin, prior to purification of the plasmids by alkaline lysis and isopropanol precipitation (*Feliciello and Chinali, 1993*). Purified plasmids were integrated into *P. tricornutum* cells via biolistic transformation, using the Biolistic PDS-1000/He Particle Delivery System (BioRad, France), essentially as previously described (*Siaut et al., 2007*; *Falciatore et al., 1999*).

Colonies obtained from each transformation were transferred to liquid f/2 supplemented with vitamins and 100 µg/ ml zeocin, and were left to recover under the same growth conditions as used for liquid cultures of untransformed cells. Expression of GFP was visualised using a TCS SP8 confocal microscope (Leica, France), an excitation wavelength of 488 nm and emission wavelength interval of c. 510–540 nm. Chlorophyll fluorescence (using an emission interval of 650–700 nm) and bright field images were simultaneously visualised for each cell. Wild-type cells that did not express GFP were used to identify the maximum exposure length possible without false detection of chlorophyll in the GFP channel (*Figure 2—figure supplement 7*).

Possible mitochondrial localisations of dual-targeted proteins were identified by staining cells with approximately 100 mM Mitotracker orange (Thermo), dissolved in filtered seawater, for 25 min under standard culture conditions (*Tanaka et al., 2015a*). Cells were rinsed and resuspended in fresh filtered seawater prior to visualisation, using the same conditions as stated above for GFP, and a 548 nm excitation laser and 575–585 nm absorbance window for the Mitotracker signal. To ensure that there was no possible crosstalk between the two signals, negative controls consisting of an unstained GFP-expressing wild-type line, and stained wild-type cells, were used respectively to determine the maximum exposure length possible without (respectively) false detection of GFP in the Mitotracker channel, and false detection of Mitotracker in the GFP channel (*Figure 7—figure supplement 1*).

## Reconstruction of evolutionary origins of ancestral plastid-targeted proteins

The most probable evolutionary origins of individual plastid-targeted proteins were identified via the combined products of BLAST top hit analysis and phylogenetic sister-group inference. First, a composite reference sequence library was generated by appending the uniref outgroup library previously used for BLAST-based assembly of ancestral HPPGs, with twenty-two combined eukaryotic transcriptome and genomic libraries of taxa with no suspected history of serial endosymbiosis, which was previously used to enrich each single-gene tree (Table S1- sheet 1 [*Dorrell et al., 2016*]). Each sequence within the library was then assigned a taxonomic affinity consisting of one of six lineages (green algae, red algae, aplastidic stramenopiles, all other eukaryotes, prokaryotes, and viruses) and one of 48 sub-categories, (Table S1- sheet 1, section 1 [*Dorrell et al., 2016*]). Next, each seed protein sequence within each ancestral HPPG was searched by BLASTp against the composite library, with a threshold e-value of $1 \times 10^{-05}$. Sequences were annotated by the lineage and sub-category of the first hit obtained, and by the number of consecutive top hits obtained within the same lineage (Table S4- sheet 2, section 2 [*Dorrell et al., 2016*]). To minimise misidentification due to any residual contamination in individual sequence libraries, only sequences for which the first three or more BLAST hits resolved within the same lineage were deemed to be unambiguously related to that lineage.

Sister-group relationships were additionally inferred for each ancestral HPPG from the previously generated single-gene trees (Table S4- sheet 2, section 3 [*Dorrell et al., 2016*]). To ensure that only true sister-group relationships were recorded, and to avoid potential misidentifications of individual sister-group relationships due to species-specific gene transfer or contaminants that had not previously been excluded by screening individual species libraries, only trees in which ochrophytes were monophyletic, (i.e., not paraphyletic with regard to any one of the five outgroups), for which a single sister-group could be identified (using the most phylogenetically complex node as the outgroup), and for which the sister-group contained at least two monophyletic or paraphyletic sequences, from different sub-categories of the same lineage, were used for subsequent analysis.

## Reconstruction of evolutionary relationships between ochrophytes and other CASH lineage plastids

To identify the probable relationships between ochrophytes and other CASH lineage plastids, each ancestral HPPG tree was enriched with sequences from six different groups of organisms with histories of serial endosymbiosis (cryptomonads, haptophytes, dinotoms, other alveolates, euglenids, and chlorarachniophytes), subdivided into thirteen sub-categories (Table S1 [*Dorrell et al., 2016*]). For the cryptomonad, haptophyte and dinotom sequences, as plastid-targeted proteins from these lineages may be identified using targeting predictors trained on diatoms such as HECTAR (*Aleoshin et al., 2016*) and ASAFind (*Gruber et al., 2015*; *Gschloessl et al., 2008*), each of the HPPGs initially generated was enriched with plastid-targeted sequences from each cryptomonad, haptophyte and dinotom sub-category identified by in silico prediction with these programmes (Table S2- sheet 1; Table S3- sheet 1 [*Dorrell et al., 2016*]).

The position of each group of organisms within the tree was then annotated as falling into one of eight different categories, four of which were internal to the ochrophytes (diatoms; hypogyristea; chrysista; or an ambiguous internal position) and four of which were external to the ochrophytes (as an immediate sister-group to all ochrophytes prior to the first outgroup lineage previously identified; within the red algae; within the green algae; and at any other position external to the ochrophytes; Table S4- sheet 2, sections 5–6 [*Dorrell et al., 2016*]). To minimise the incorporation of contaminant and non-plastid sequences, tree positions were only recorded if the branch containing sequences from that particular lineage included at least two of the sub-categories considered (for alveolates, cryptomonads, and haptophytes), contained at least one predicted plastid-targeted sequence (for dinotoms, cryptomonads and haptophytes), and for which only one category could be applied (i.e., the tree only contained one evolutionarily distinct group for each lineage, which could be unambiguously allocated one category over all others). Each tree annotation was repeated three times independently, and only tree annotations that were recorded consistently in each case were retained for further analysis.

To identify plastid-targeted proteins that were uniquely shared between haptophytes and other lineages, every HPPG initially generated was screened for the inclusion of only two of five different lineages (diatoms including dinotoms, hypogyristea, chrysista, haptophytes, and cryptomonads; Table S2- sheet 2, section 3; Table S3- sheet 2, section 3 [*Dorrell et al., 2016*]). The frequencies of these proteins were then compared to the numbers expected in a random distribution of all uniquely shared HPPGs across the entire dataset: for example, if half of all uniquely shared HPPGs were shared with diatoms and one other lineage, and half were shared with haptophytes and one other lineage, then one-quarter of all uniquely shared HPPGs should be shared between haptophytes and diatoms. The expected numbers were corrected to take account of the expected frequencies calculated through this approach to be uniquely shared within one lineage only: for example, in the above case, one-quarter of the expected frequency would be allocated to HPPGs uniquely present in diatoms; to correct for this, all remaining expected frequencies of uniquely shared HPPGs would therefore be multiplied by four-thirds (i.e. one-third of all uniquely shared HPPGs should be shared between haptophytes and diatoms).

The specific evolutionary relationships associated with haptophyte plastid-targeted proteins incorporated into ancestral HPPGs were investigated using a modified BLAST top hit technique. Firstly, all of the plastid-targeted proteins assembled into each ancestral HPPG were extracted and separated into each separate sub-category (Table S13- sheet 1 [*Dorrell et al., 2016*]). Each sub-category list was then reduced to only leave one, randomly selected sequence per HPPG (Table S13- sheet 2 [*Dorrell et al., 2016*]). Finally, each sequence retained in the reduced list was searched by BLAST against a composite library, consisting of the library previously used for outgroup top hit analysis, enriched with all of the plastid-targeted proteins identified for ochrophytes, haptophytes and cryptomonads, except for those that corresponded to the same particular lineage as the query sequence (Table S13- sheets 1,3 [*Dorrell et al., 2016*]). For example, in the case of haptophytes, plastid-targeted sequences that had been separated into three individual categories (pavlovophytes, prymnesiales, and isochrysidales [*Simon et al., 2013*]) were searched against a composite library consisting of all outgroup sequences, and plastid-targeted sequences from diatoms, hypogyristea, chrysista, and cryptomonads, but excluding haptophytes. BLAST top hit analysis was then performed as described above (Table S13- sheets 1, 3 [*Dorrell et al., 2016*]). Finally, to enable the identification of genes with consistent results from multiple analyses, the lineage of the BLAST top hit was compared to the lineage of the haptophyte sister-group in the single-gene tree analysis (Table S4- sheet 2, section 5; Table S13- sheet 4 [*Dorrell et al., 2016*]).

## Identification of uniquely shared residues in multigene HPPG datasets

To identify residues that are uniquely shared between ochrophytes and other lineages, multigene datasets were constructed of a) ancestral HPPGs of green algal origin, and b) ancestral HPPGs for which haptophytes show origins within the ochrophytes. To minimise the incorporation of sequences of misidentified origin, in each case only the HPPGs for which the proposed evolutionary origin were identified both by BLAST top hit and single-gene tree analysis were included. To avoid introducing artifacts due to lineage-specific gene transfers, paralogy events, or other phylogenetic incongruencies that could otherwise bias the eventual results (*Qiu et al., 2012*; *Leigh et al., 2008*), the single-gene tree generated for each HPPG was manually inspected to exclude any that contain multiple clades (defined as monophyletic groups containing more than one sequence from a particular lineage, separated from one another by at least two sequences from outside that particular lineage) for each of the major lineages of interest within the tree:

- For the green gene dataset, HPPG trees containing more than one clade of ochrophyte, cryptomonad, haptophyte, red algal, or green algal sequences were excluded. To account for the possibility that CASH lineage sequences might originate from within the green algae, the green algae were allowed to be paraphyletic with regard to the cryptomonad, haptophyte and ochrophyte sequences, but were not allowed to incorporate sequences from other lineages. Similarly, to account for the possibility that subsequent gene transfers may have occurred from ochrophytes into other CASH lineages, the ochrophytes were allowed to be paraphyletic with regard to cryptomonad and haptophyte sequences, but not to any other lineages.
- For the haptophyte gene dataset, HPPG trees containing more than one clade of ochrophyte, haptophyte, diatom, hypogyristean, or chrysistan sequences were excluded. To account for the possibility that haptophytes arose within the ochrophytes, the ochrophyte, diatom,

hypogyristean and chrysistan sequences were allowed to incorporate sequences from hapto-phytes. Similarly, due to the paraphyly of hypogyristea with regard to diatoms, the hypogyris-tean sequences were allowed to incorporate sequences from diatoms, but not from other lineages.

- In all cases, sequences from chlorarachniophytes, euglenids, and alveolates were not incorpo-rated into any of the clade assessments, due to uncertainty over the gene transfer events that have occurred in each lineage (*Ševčíková et al., 2015*; *Maruyama et al., 2011*; *Archibald et al., 2003*).

This left datasets consisting of 32 HPPGs for which the ochrophytes were of clear green algal ori-gin, and 37 HPPGs in which the haptophytes were of clear ochrophyte origin, with no conflicting phylogenetic signal. The rationale for inclusion and exclusion of each HPPG in each analysis is pre-sented in Table S6, sheets 1 and 3 (*Dorrell et al., 2016*).

Next, to eliminate individual sequences remaining within each HPPG that might have arisen through species-specific gene transfer or contamination events, each trimmed sequence within each approved alignment was inspected using a composite BLAST approach. First, each sequence was searched against a composite library containing all uniref, jgi and MMETSP sequences from every lineage within the tree of life, and the top ten hits were tabulated for each sequence. In each case, only sequences for which at least the first three hits were of the same lineage as that of the query were retained. For the haptophyte multigene alignment, the ochrophytes were separately analysed as each of the three component lineages (chrysista, hypogyristea, and diatoms), which is to say that a query obtained from a member of the hypogyristea would only be retained if the first three BLAST top hits originated from other hypogyristean sequences, rather than other ochrophytes.

Next, each of the component sequences within each cleaned alignment were searched against all other component sequences within the same alignment using BLASTp, and the top ten hits within the alignment were ranked. In each case, sequences were only approved for incorporation into the multigene dataset if the first non-self hit was to a different sub-category within the same lineage, e.g. if a query sequence from a red alga yielded a top hit against a red algal sequence from a differ-ent red sub-category. To allow for possible cases of paraphyly and/or absence of sequences within each alignment, the following modifications were applied:

- Green algal sequences within the confirmed green origin alignments were allowed to yield top hits against ochrophytes, cryptomonads, and haptophytes, but were required to yield a best hit against another green alga with an expect value lower than the top hit against red algal or glaucophyte sequences.
- Glaucophyte sequences were deemed to be of correct origin if they yielded a top hit against cyanobacteria, red algae, or green algae, due to the incorporation (in general) of only one glaucophyte sequence in each alignment.
- Ochrophyte sequences were deemed to be of correct origin if they yielded a top hit against any other ochrophyte sub-category (regardless of whether this was of diatom, hypogyristean or chrysistan origin). Ochrophyte sequences were additionally allowed to yield top hits against cryptomonads (in the green gene alignments), and haptophytes (in both green and hapto-phyte gene alignments), but were required to yield a best hit against another ochrophyte with an expect value lower than the best hit against green algal, red algal or glaucophyte sequences.
- Sequences for which no top hits were found for a different sub-category within the same line-age, but for which at least one top hit were found within the same sub-category within the lineage, and for which the first ten BLAST hits did not directly indicate a contamination event, were deemed to be of correct origin.

Tabulated outputs for each BLAST analysis are provided in Table S6, sheets 2 and 4. Finally, each dataset was reduced to leave only one randomly selected sequence for each given sub-category within each HPPG alignment.

The number of residues that were uniquely shared between ochrophytes and green algae in the green gene dataset, and haptophytes and ochrophytes in the haptophyte dataset, were then tabu-lated (Table S7 [*Dorrell et al., 2016*]). Briefly, residues were inferred to be uniquely shared between ochrophytes and green algae if they were present in at least 2/3 of the ungapped ochrophyte sequences, one or more green algal sequence, and if none of the red algal or glaucophyte sequen-ces shared the residue in question, but at least one of these sequences had a non-matching (i.e.

non-gapped) residue at that position (Table S7- sheet 1, section 2 [*Dorrell et al., 2016*]). Similarly, residues were inferred to be uniquely shared between ochrophytes and haptophytes if they were present in at least 2/3 of the ungapped haptophyte sequences, one or more ochrophyte sequence, and if none of the green algal, red algal, glaucophyte or cyanobacterial sequences shared the residue in question, but at least one of these sequences had a non-matching (i.e., non-gapped) residue at that position (Table S7- sheet 2, section 2 [*Dorrell et al., 2016*]). The origin point of each uniquely shared residue was then inferred by comparison to reference topologies respectively of green algae (*Leliaert et al., 2011*) and of ochrophytes (per *Figure 1*). Residues were assumed to have originated in a common ancestor of a particular clade if that clade contained more lineages with matching than non-matching or gapped residues (Table S7- sheets 1–2, section 5 [*Dorrell et al., 2016*]). A second analysis was additionally performed in which all gapped residues were deemed to be matching, to identify the earliest possible origin point for each uniquely shared residue, taking into account secondary loss (*Ku et al., 2015*; *Qiu et al., 2015*) and absence of sequences from each alignment (*Woehle et al., 2011*; *Deschamps and Moreira, 2012*).

## Analysis of targeting preferences of ancestral ochrophyte and haptophyte genes

Two libraries of non-redundant gene families that were broadly conserved across ochrophytes or haptophytes, and thus might represent gene products of the ancestral genomes of these lineages, were generated using a similar BLAST-based assembly pipeline as used to construct HPPGs (Table S8; Table S14 [*Dorrell et al., 2016*]). Ochrophyte gene families were deemed to be conserved if orthologues were detected in one of three different patterns of ochrophyte sub-categories previously defined to correspond to ancestral plastid-targeted proteins (*Figure 1*, panel B; Table S8- sheet 1, section 3 [*Dorrell et al., 2016*]). Haptophyte gene families, built through a similar pipeline using seed sequences from the *Chrysochromulina tobin* and *Emiliania huxleyi* genomes (*Read et al., 2013*; *Hovde et al., 2015*), were deemed to be ancestral if orthologues were identified in at least two of the three haptophyte sub-categories considered (pavlovophytes, prymnesiales, and isochrysidales; Table S14- sheet 1, section 3 [*Dorrell et al., 2016*]).

The most probable evolutionary origin of each gene family was inferred by BLAST top hit analysis of the seed sequence (Table S8- sheets 1, 2; Table S14- sheets 1, 2 [*Dorrell et al., 2016*]). Ochrophyte sequences were searched against the composite uniref + MMETSP library used to previously identify the most likely outgroup to each ancestral plastid-targeted protein (Table S8- sheet 1, section 6 [*Dorrell et al., 2016*]), while haptophyte sequences were searched against the enriched library that also contained all ochrophyte and cryptomonad sequences, to enable the distinction of proteins of probable CASH lineage plastid origin from proteins that had evolved through independent gene transfer events between haptophytes and non-CASH lineage organisms (Table S14- sheet 1, section 6 [*Dorrell et al., 2016*]). Targeting preferences for each protein encoded within each gene family were identified using SignalP v 3.0 and ASAFind v 2.0 (*Dorrell et al., 2016*), and with HECTAR (*Gschloessl et al., 2008*), as previously discussed (Table S8- sheet 3; Table S14- sheet 3 [*Dorrell et al., 2016*]). Targeting preferences that were identified in a plurality of sequences and in $\geq$2/3 of the sequences within each ochrophyte gene family were recorded (Table S8- sheet 2, sections 4–5 [*Dorrell et al., 2016*]). As only three haptophyte sequences were assembled for each ancestral haptophyte gene family, only targeting predictions that were identified in $\geq$2/3 of the sequences within the HPPG were inferred to be genuine (Table S14- sheet 2, sections 4–5 (*Dorrell et al., 2016*]).

## Functional and physiological annotation of ancestral plastid-targeted proteins

Core plastid metabolism pathways were identified using recent reviews of ochrophyte metabolism, or reviews of homologous plant plastid metabolic pathways where ochrophyte-specific reviews have not yet been published (*Smith et al., 2012*; *Green, 2011*; *Grouneva et al., 2011*; *Allen et al., 2011*; *Kroth et al., 2008*; *Bromke, 2013*; *Bertrand, 2010*; *Miret and Munné-Bosch, 2014*; *Bandyopadhyay et al., 2008*; *Shtaida et al., 2015*). The probable function and KOG classification of each HPPG were annotated using the pre-existing annotations associated with seed protein sequence (if these existed), or if not the annotated function of the top uniref hit previously identified

by BLAST searches of the seed sequence (Table S9 [*Dorrell et al., 2016*]). Expression dynamics for each ancestral HPPG within the genomes of the model diatoms *Phaeodactylum tricornutum* and *Thalassiosira pseudonana* were inferred using microarray data integrated into the DiatomPortal server (*Ashworth et al., 2016*) (Table S10- sheets 1,2 [*Dorrell et al., 2016*]). Correlation coefficients were calculated between each pair of *P. tricornutum* and *T. pseudonana* genes that were incorporated into an ancestral HPPG, across all microarray libraries within the dataset (Table S10- sheets 3,4 [*Dorrell et al., 2016*]), with average values being calculated from all pairwise correlations for different evolutionary categories of protein (Table S10- sheet 5 [*Dorrell et al., 2016*]).

Possible chimeric proteins, resulting from the fusion of proteins of different evolutionary origins, were identified in the dataset using a modified version of a previously published protocol (*Méheust et al., 2016*) (Table S9- sheet 1, sections 4,5; Table S11 [*Dorrell et al., 2016*]). Each protein within each HPPG was searched using BLASTp against the composite outgroup MMETSP-enriched library, using the same taxonomic classification used for the identification of the evolutionary origin of each seed protein within the dataset, and all hits with an expect value of $1 \times 10^{-05}$. Component sequences were then grouped into component families according to the following rule: if two component sequences overlapped by more than 70% of their lengths on the protein composite, they belonged to the same component family. Overlapping and/ or nested component families were additionally merged if one family was included by more than 70% of its length into the other one. Component families were then assigned a broad evolutionary origin corresponding to their taxonomic composition. If the three best component sequences, according to their BLAST bitscore against the composite gene, matched with the same lineage (e.g., green algae, red algae, aplastidic stramenopiles, or other eukaryotes), the component was considered to have originated from that lineage.

Possible dual-targeted proteins were identified within the dataset by screening all possible plastid-targeted proteins with Mitofates, using a cut-off targeting threshold of 0.35 (*Fukasawa et al., 2015*), which was inferred to be more effective in identifying experimentally verified ochrophyte mitochondria-targeted proteins (*Figure 7—figure supplement 2*) (*Gruber et al., 2015*) than other threshold values or targeting prediction programmes such as TargetP (*Emanuelsson et al., 2007*) or Mitoprot (*Claros, 1995*). The default Mitofates positive cutoff value was modified from 0.38 to 0.35 in order to maximise the capture of experimentally localised mitochondrial proteins, without admitting proteins with unambiguous plastid localisation (*Figure 7—figure supplement 2*). As dual-targeting to plastids and mitochondria may be achieved either by distinct protein isoforms resulting from ambiguous targeting peptides or alternative internal translation initiation sites that allow production of mitochondrial targeting sequences (*Xu et al., 2013*; *Hirakawa et al., 2012*), each protein was screened with Mitofates using both the full-length N-termini, and N-termini predicted to result from the next downstream methionine within 30 residues. Possible conserved dual-targeted proteins were then identified via the same BLAST-based assembly pipeline and stringency thresholds used to identify probable ancestral HPPGs (Table S12- sheet 1 [*Dorrell et al., 2016*]). All putative dual-targeted proteins have been made publically accessible through the University of Cambridge dSpace server (https://www.repository.cam.ac.uk/handle/1810/261421) (*Dorrell et al., 2016*).

## Construction and inspection of concatenated and exemplar phylogenetic trees

For the plastid genome phylogenetic analysis, single-gene alignments were constructed by BLAST searches of published red lineage and glaucophyte plastid genomes (for the gene rich analysis) or of these genomes plus all MMETSP libraries for the same lineages (for the taxon rich analysis), using the *Phaeodactylum tricornutum* protein sequence as query and a threshold e-value of $1 \times 10^{-05}$, followed by alignment using GeneIOUS v 4.76 (*Kearse et al., 2012*), as before. The gene rich analysis included protein sequences from 54 genes that were identified in 22 different non-green lineage plastid genomes while the taxon-rich analysis included 10 different plastid genes that were identified in all 22 plastid genomes and at least 30 different MMETSP libraries (*Keeling et al., 2014*) (Table S15- sheet 1 [*Dorrell et al., 2016*]). For the taxon-rich analysis, only species that were represented in ≥6/12 of the single-gene alignments were included in the concatenated alignment. Each concatenated alignment was trimmed using trimal (*Capella-Gutiérrez et al., 2009*) using the -gt 0.8 option.

Single-gene alignments for four plastid-targeted proteins predicted to be of polyphyletic origin in ochrophytes (3-dehydroquinate synthase, isopropylmalate dehydratase, sedoheptulose bisphosphatase, and shikimate kinase) were generated using a similar BLAST-based assembly and alignment pipeline as used to verify ancestral plastid-targeted proteins. In this case, all non-redundant (as inferred by BLAST top hit evalue) plastid-targeted sequences for each protein identified from ochrophyte genomes were used as independent queries for the identification of plastid-targeted orthologues, 50 uniref top hits, and top hits from the combined MMETSP and genomic libraries from 36 eukaryotic sub-categories, as before. HPPGs were independently generated, aligned and trimmed for each seed sequence; all HPPGs generated for each protein were then merged, realigned and retrimmed using trimAl to generate a single-gene alignment. Single-gene alignments for each of the constituent genes in each concatenated plastid genome tree were generated by splitting the alignment into its component genes. All alignments have been made publically accessible through the University of Cambridge dSpace server (https://www.repository.cam.ac.uk/handle/1810/261421) (*Dorrell et al., 2016*).

Trees were inferred for each concatenated and exemplar single-gene alignment (Table S15- sheet 2 [*Dorrell et al., 2016*]) using the MrBayes and RAxML programmes in-built into the CIPRES webserver (*Stamatakis, 2014*; *Miller et al., 2015*; *Ronquist et al., 2012*). Bayesian trees were inferred using three substitution models (GTR, Jones, and WAG), a minimum of 600000 generations, and an initial burn-in discard value of 0.5. Trees were only utilised if the final convergence statistic between the two chains run was ≤0.1, and tree calculation was automatically stopped if the convergence statistic fell below 0.01. RAxML trees were inferred using three substitution models (GTR, JTT, and WAG) with automatic bootstopping, as previously described (*Dorrell et al., 2017*). The best tree topology for each RAxML tree was inferred, and bootstrapping was performed using a burnin value of 0.03. Alternative tree topologies were tested for the RAxML + JTT tree inferred from each concatenated alignment using CONSEL (*Shimodaira and Hasegawa, 2001*), under the default conditions. Tree outputs have been made publically accessible through the University of Cambridge dSpace server (https://www.repository.cam.ac.uk/handle/1810/261421) (*Dorrell et al., 2016*).

Modified alignments were generated for both of the plastid concatenated multigene datasets from which individual clades of organisms (diatoms, hypogyristea, chrysista, haptophytes, cryptomonads, red algae, and different combinations of green algae) had been removed (Table S15- sheet 2 [*Dorrell et al., 2016*]). Fast-site removal was performed using TIGER (*Cummins and McInerney, 2011*). Site rate evolution characteristics were calculated for each alignment using the -b 100 option, and modified alignments were constructed from which the rate categories corresponding to the fastest evolving 40–50% of sites were serially removed (Table S15- sheet 2 [*Dorrell et al., 2016*]). Amino acid composition for each plastid alignment were calculated, and two modified alignments were generated from which glycines (which in all alignments occur at significantly lower frequencies in ochrophytes than in haptophytes or cryptomonads; chi-squared, p≤0.05; Table S16- sheet 3 [*Dorrell et al., 2016*]), and from which seven amino acids (alanine, aspartate, glycine, histidine, leucine, asparagine, threonine and valine) which were found in at least one alignment to occur at significantly different frequencies in ochrophytes compared to haptophytes or to cryptomonads (p≤0.05; Table S16- sheet 3 [*Dorrell et al., 2016*]) had been removed. Trees were inferred for each modified alignment using RAxML with the JTT substitution, and MrBayes with the Jones substitution, and bootstrap calculation as previously described. Modified alignments and tree outputs have been made publically accessible through the University of Cambridge dSpace server (https://www.repository.cam.ac.uk/handle/1810/261421) (*Dorrell et al., 2016*).

Uniquely shared residues were manually tabulated for both of the plastid genome multigene alignments (Table S17 [*Dorrell et al., 2016*]). For the gene-rich plastid multigene alignment, residues that were present in all haptophyte sequences and only found in a maximum of one other lineage (red algae, glaucophytes, cryptomonads, diatoms, hypogyristea, or chrysista) were tabulated (Table S17- sheet 1 [*Dorrell et al., 2016*]). For the taxon-rich alignment, to take into account gaps and missing characters, residues were tabulated if they were found in a majority of haptophyte sequences, and one other lineage, as before (Table S17- sheet 2 [*Dorrell et al., 2016*]). The total number of residues shared, and uniquely shared, with each non-haptophyte species and lineage are respectively tabulated in Table S17, sheets 3 and 4 (*Dorrell et al., 2016*).

## Data deposition

All supporting datasets for this study, including supplementary tables predicted plastid-targeted and dual-targeted protein libraries, single gene and multigene alignments, and tree outputs, have been made publically and freely accessible through the University of Cambridge dSpace server (https://www.repository.cam.ac.uk/handle/1810/261421) (*Dorrell et al., 2016*).

## Acknowledgements

The authors would like to thank Achal Rastogi (École Normale Supérieure), Neal Clarke (Yale University), Michael Melkonian (University of Koln), Gane Ka-Shu Wong (University of Alberta) and Jun Yu (Beijing Institute of Genomics) for early access to sequence data used in this study, and Catherine Cantrel, Anne-Flore Deton-Cabanillas, Zhanru Shao, Leïla Tirichine and Javier Paz-Yepes (École Normale Supérieure) for assistance with generation of transgenic expression constructs for *Phaeodactylum tricornutum*. Funding is acknowledged from the ERC Advanced Award ''Diatomite'', the Louis D Foundation of the Institut de France, the French Government ''Investissements d'Avenir'' programmes MEMO LIFE (ANR- 10-LABX-54) and PSL* Research University (ANR-11-IDEX-0001–02), and the Gordon and Betty Moore Foundation (all to CB), and from FP7 (grant number 2007–2013 Grant Agreement 615274, to EPB). RGD is supported by an EMBO early career fellowship (ALTF 1124–2014). DJR was supported by a postdoctoral fellowship from the Conseil Régional de Bretagne and the French Government 'Investissements d'Avenir' programme OCEANOMICS (ANR-11-BTBR-0008). CB was the Grass Fellow at the Radcliffe Institute for Advanced Study, Harvard University during the 2016-2017 academic year. The authors would like to thank the handling reviewer and two anonymous editors for constructive comments on the manuscript.

## Additional information

### Funding

| Funder | Grant reference number | Author |
| --- | --- | --- |
| EMBO | ALTF 1124/2014 | Richard G Dorrell |
| European Research Council | Diatomite | Chris Bowler |
| Institut de France | Louis-D Foundation | Chris Bowler |
| Seventh Framework Programme | 615274 | Eric P Bapteste |
| Gordon and Betty Moore Foundation | | Chris Bowler |
| Paris Sciences et Lettres, Labex-MemoLife | ANR- 10-LABX-54 | Chris Bowler |
| Agence Nationale de la Recherche | ANR-11-IDEX-0001-02 | Chris Bowler |
| Agence Nationale de la Recherche | ANR-11-BTBR-0008 | Daniel J Richter |

The funders had no role in study design, data collection and interpretation, or the decision to submit the work for publication.

### Author contributions

RGD, Conceptualization, Data curation, Formal analysis, Supervision, Funding acquisition, Validation, Investigation, Visualization, Methodology, Writing—original draft, Project administration, Writing—review and editing; GG, Formal analysis, Supervision, Investigation, Methodology, Writing—review and editing; GM, Formal analysis, Validation, Investigation; RM, Data curation, Formal analysis, Investigation, Methodology; EPB, Supervision, Methodology, Writing—review and editing; CMK, Formal analysis, Investigation; LB-G, Resources, Software, Methodology; KDF, Investigation, Methodology; DJR, Resources, Data curation, Software; CB, Conceptualization, Supervision, Funding acquisition, Writing—review and editing

## Author ORCIDs

Richard G Dorrell, http://orcid.org/0000-0001-6263-9115

Raphaël Méheust, http://orcid.org/0000-0002-4847-426X

Daniel J Richter, http://orcid.org/0000-0002-9238-5571

Chris Bowler, http://orcid.org/0000-0003-3835-6187

## Additional files

### Major datasets

The following dataset was generated:

| Author(s) | Year | Dataset title | Dataset URL | Database, license, and accessibility information |
|---|---|---|---|---|
| Dorrell RG, Gile G, McCallum G, Guegen L, Klinger CM, Meheust R, Peterson K, et al | 2016 | Research data supporting "The ancestral ochrophyte plastid proteome" | https://www.repository.cam.ac.uk/handle/1810/261421 | Publicly available at the University of Cambridge Repository - Apollo |

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
