## [Decision Letter]

Thank you for submitting your article "Reconstruction of an ancestral plastid proteome reveals the origins and fate of complex plastid lineages" for consideration by *eLife*. Your article has been reviewed by two peer reviewers, and the evaluation has been overseen by a Reviewing Editor and Diethard Tautz as the Senior Editor. The reviewers have opted to remain anonymous.

The reviewers have discussed the reviews with one another and the Reviewing Editor has drafted this decision to help you prepare a revised submission.

Summary:

Dorrell et al. use combination of genomic and cell biology methods to study the origin and evolution of the plastid proteome in a key group of phytoplankton and protists (i.e., both photosynthetic and aplastidial) that comprise the so-called CASH lineage (photosynthetic members of the cryptomonads, alveolates, stramenopiles, and haptophytes). These taxa are major contributors to oceanic primary production and have a 'checkered past' with respect to how scientists have explained the origin and spread of their plastids through endosymbiosis. Many questions remain about how many endosymbioses explain the origin of CASH plastids and if some lineages have undergone cryptic endosymbioses involving other algae. These latter events leave putative genetic 'footprints' in the nuclear genome that is explained by ancient endosymbiotic gene transfer (EGT). With the arrival of the most recent plastid of a different phylogenetic origin, a subsequent round of EGT is initiated. Therefore, CASH genomes have complex evolutionary histories.

Specifically, the authors explored the question: which nuclear-encoded plastid-targeted proteins have had the most fundamental roles in plastid establishment? Briefly, they developed an in silico pipeline to characterize the ancestral ochrophyte (photosynthetic stramenopiles) plastid-targeted proteins from 11 ochrophyte genomes and 157 ochrophyte transcriptomes, and verified plastid localization for a few unusual candidate proteins experimentally through GFP fusion constructs. Then, they used the list of presumptive ancestral ochrophyte plastid-targeted proteins to reconstruct metabolic pathways in the ancestral ochrophyte plastid, determined the phylogenetic origins of these plastid-targeted proteins, deduced a late acquisition of the ochrophyte plastid (i.e., independent from other CASH lineages), found support for the previously observed (but debated) green lineage contribution to the ochrophyte plastid proteome, correlated metabolic functions of proteins to their phylogenetic origins, identified phylogenetically chimeric plastid-targeted proteins, explored plastid/mitochondrion interactions in the ochrophytes (including dual-targeting), and observed a strong pelagophyte contribution to the haptophyte plastid proteome which they interpret as remnants from a preceding pelagophyte endosymbiont/plastid. Both reviewers felt that, whereas this comprehensive collection of data and analyses had great merit in explaining key transitions in plastid evolution, that in its current form, the manuscript did not adequately describe some aspects of the work. Improving the focus and clarity, particularly in respect to figures, are critical for anyone from outside the field to understand this good but very complex manuscript.

These shortcomings are listed below.

Essential revisions:

Focus: The authors should focus the text to make it clearer for the reader to understand. Emphasis statements are needed so that the reader can evaluate the significance of a piece of information over another. Well-supported sections are interspersed with those that are weak and the authors jump between evolution and metabolism several times, which is quite confusing. For example, the first paragraph in the second section of Results and all of section 4 in the Results are relatively less interesting. Dual-targeting is neither novel nor fundamentally relevant to the scope of this paper, despite the value of the work that was done. Other examples are the first section of results of *rpl36*, which is inconclusive and cannot reject with confidence the possibility of contamination in the data.

Paper title: A very general and rather unclear title. Perhaps the term "ochrophytes" should be used and the latter part deleted – the authors do not address the fate of other complex plastids except for those in haptophytes, which is a small section of the paper and one that is intimately linked to ochrophytes.

Concatenated datasets (nuclear): This referee was skeptical about concatenating single gene trees with "similar" signals, such as "green" and "red" trees (subsection “A significant green algal contribution to ochrophyte plastid evolution”, third paragraph) and haptophyte genes (see below). The evolutionary signal in nuclear plastid-targeted genes is notoriously complex. This is one reason why one has to be stringent when it comes to selecting genes that are interpretable in this context. By concatenation, the assumption is that the topologies are congruent across all genes. This is unlikely and if it is the case then tests for topology congruence should be done, also using manual inspection of the data.

Contamination: First, the fact that this was done should be more clearly presented (in Results) because the extent of contamination is quite serious, particularly in MMETSP. Second, removing 100% identical pairwise hits is not stringent enough, strictly speaking. No mismatches above 95% or 97% should be used when comparing distantly related groups (this would not work for closely related species) because errors and un-removed sequencing adapters often show up as differences. Also, the screen for contaminations should ideally have been done at the nucleotide level. The authors do not need to redo the analysis, however it is important that it is clearly stated in the text that further controls downstream (e.g., careful manual evaluation of each single tree) would have identified any remaining contamination and not bias the results.

Cryptic endosymbioses: The foreign gene signals (green prasinophyte and haptophyte/pelagophyte) that were identified are likely to be genuine. These results are viewed as being most consistent with cryptic endosymbioses, which is controversial primarily because the assumptions rely on weakly supported precedents (extent of EGT in secondary endosymbiosis) or assumptions where essentially no evidence exists (e.g., strong HGT from a single source is unlikely). Therefore, please keep the discussion about cryptic endosymbioses as provisional.

Figures and figure references: The figure references are unclear and not in a uniform format. Here are several examples:

"(Figure 2—figure supplement 1; Table S6)" vs. "(Figure 2, panel C; Figure 2—figure supplement 2, Figure 2—figure supplement 3 – Table S5)"

"(Figures S6-S8)" Where are these figures?

Within the figure legend of Figure 6—figure supplement 2: "(Figure 5; Figure S25)". Where is Figure S25?

Within Figure 2—figure supplement 2: why does it say "Figure S4" next to panel H?

The figures are also unclear and the descriptions insufficient. Here are a few examples:

Figure 2, panel A: what does the grey and white color codes indicate? The legend shows violet color that does not occur in the figure. Panel B: black arrows point in between categories.

Figure 2—figure supplement 2: the figure stretches over 4 pages and is very inconvenient to look at, the pathways are difficult to recognize and the referee cannot find explanations for the abbreviations used, the color code used is unclear (e.g.: What does the dark blue-green box used for the Kynurenine aminotransferase represent? What do blue or turquois letters as opposed to black letters signify?).

Figure 3: Panel B, and C, and Figure 3—figure supplements 2-5: The font used in the various figures is too small. Text within alignments and trees is illegible. This referee could not find a legend to the color code used to assign taxa within phylogenetic trees to specific lineages.

Supplements: There is an excessive use of supplemental tables in this manuscript. Many of these tables are cryptic and the interesting information is buried in them instead of being explained in the text. Here is one example:

"We additionally noted that within our dataset there were several HPPGs which have been previously annotated to function in organelles other than the plastid (for example, twenty HPPGs of annotated mitochondrial function, and four each of annotated endosomal and peroxisomal functions; Table S5). […] Other proteins, which lack any obvious homologues within plastid proteome data, might contribute entirely novel biological functions to the ochrophyte plastid proteome (Tables S4, S5)." What type of proteins are we looking at here and which novel biological functions could they contribute?

Main text: Not all claims made in the text are sufficiently supported by the data. Here are several examples:

The referee would like to see wild type controls for the fluorescence microscopic images.

"In a majority (304/551) of trees for which plastid-targeted haptophyte homologues could be identified, haptophyte orthologues resolved within the ochrophytes (Figure 7, panel A). All other groups considered, including the alveolates, which have been proposed to possess a plastid of ochrophyte origin, more frequently branched external to than within the ochrophytes (Figure 7, panel A)." The text suggests that Figure 7, panel A resolves the phylogenetic positions of proteins within the ochrophytes which is not the case.

"…However, these resolved as a separate clade to the C-type haptophyte and cryptomonad plastid *rpl36* sequences, which formed a monophyletic clade with consistent support (Figure 8, panel D; consensus support > 0.5 in all Bayesian analyses), and were unified by three uniquely shared residues that were absent from all ochrophyte sequences (Figure 8, panel D)." The referee did not see a monophyletic clade of haptophyte and cryptomonad *rpl36* sequences in the tree. The tree does not resolve the position of haptophytes/cryptomonads relative to dinoflagellates. Furthermore, a Bayesian probability >0.5 cannot be considered as consistent support. Finally, one of the three "uniquely shared" residues in the alignment that unifies the haptophytes is not shared by Pavlova.

---

## [Author Response]

Essential revisions:

Focus: The authors should focus the text to make it clearer for the reader to understand. Emphasis statements are needed so that the reader can evaluate the significance of a piece of information over another.

We have revised each Results section so that it states the context and biological rationale first. For example, in the context of exploring the plastid functions associated with the ochrophyte ancestor, we have clarified that:

" It has previously been speculated, for example, that genes acquired by diatoms from green algae might have a specific role in tolerating variable light regimes^1-3^ or eliminating toxic substances from diatom plastids^4^….[in] a converse scenario [the] mosaic origins of the ochrophyte plastid have led to the functional mixing of enzymes with disparate evolutionary origins"

Well-supported sections are interspersed with those that are weak and the authors jump between evolution and metabolism several times, which is quite confusing. For example, the first paragraph in the second section of Results and all of section 4 in the Results are relatively less interesting.

We have merged the second and fourth section of the Results, presenting a single, streamlined exploration of the biochemical functions associated with the ancestral ochrophyte plastid. We have restricted the functional analysis to techniques where we have experimental confirmation (dual-targeting), parallel data from multiple species (coregulation of different evolutionary categories of plastid genes), or particularly clear or striking evidence of differences between datasets (e.g. ochrophyte-specific fusion proteins, and multiple differences in the proportions of different KOG annotations in the ancestral HPPG dataset versus the full dataset). We have excluded discussions of GO annotations (which are dependent on rather small effective sample sizes), coregulation of Phaeodactylum genes to the mitochondrial genome (as this is based on data from only one species), and phylogenies of genes with predicted overlapping functions (which really requires experimental validation of the expression dynamics and kinetics of each gene family) from the revised manuscript.

Dual targeting is neither novel nor fundamentally relevant to the scope of this paper, despite the value of the work that was done.

We feel that dual-targeting provides valuable insights into plastid establishment. The incoming plastid proteome clearly acquires proteins from various genetic sources, but did it also modify the proteome of other organelles? In our opinion both perspectives are essential for a holistic view of plastid establishment. We have clarified this directly in the text that the aim of this section is to assess "whether the acquisition of the ochrophyte plastid might have also fundamentally altered the biology of the host cell". As we are purely presenting this as an evolutionary, rather than functional, question we have removed all discussion of protein function (e.g. KOG annotation) from this section.

Other examples are the first section of results of rpl36, which is inconclusive and cannot reject with confidence the possibility of contamination in the data.

Indeed, because the *rpl36* data that we presented is merely confirmatory to the results presented by Rice and Palmer in 2006 (i.e. we can find no additional conclusive evidence that any ochrophyte plastid genome contains a haptophyte-type *rpl36* gene), and because we cannot be sure that the *rpl36* sequences present in MMETSP are of plastome origin (as opposed to being from bacterial contaminants in each library), we have removed *rpl36* from the Results section. We have chosen instead to briefly state in the Discussion that the data presented by Rice and Palmer confirms the dissimilarity of ochrophyte and haptophyte plastid genomes, alongside other previously published synapomorphies and differences that support the results in this study (Discussion, fifth paragraph).

Paper title: A very general and rather unclear title. Perhaps the term "ochrophytes" should be used and the latter part deleted – the authors do not address the fate of other complex plastids except for those in haptophytes, which is a small section of the paper and one that is intimately linked to ochrophytes.

We have accordingly modified the title to "chimeric origins of ochrophytes and haptophytes revealed through an ancient plastid proteome", which places the focus on lineages explored.

Concatenated datasets (nuclear): This referee was skeptical about concatenating single gene trees with "similar" signals, such as "green" and "red" trees (subsection “A significant green algal contribution to ochrophyte plastid evolution”, third paragraph) and haptophyte genes (see below). The evolutionary signal in nuclear plastid-targeted genes is notoriously complex. This is one reason why one has to be stringent when it comes to selecting genes that are interpretable in this context. By concatenation, the assumption is that the topologies are congruent across all genes. This is unlikely and if it is the case then tests for topology congruence should be done, also using manual inspection of the data.

We have removed both the concatenated analyses from the manuscript. Instead, to place the evolutionary origins of the green genes in ochrophytes, and the ochrophyte genes in haptophytes in a broader phylogenetic context, we perform two additional analyses that resolve the deeper phylogenetic affinities of each set of genes without reliance on concatenated analysis. These analyses have been designed to overcome the limitations of reliance on individual gene trees and BLAST top hits to recover evolutionary origins – namely that the signal from individual gene trees is likely to be unreliable and biased by instances of secondary gene loss and lateral gene transfer, and without profiling the deeper phylogenetic affinities associated with each sister-group relationship it is not possible to determine the directions in which each gene transfer occurred.

First, we compare the specific sister-groups associated with the ochrophyte genes of green origin, and the haptophyte genes of ochrophyte origin to published phylogenies of green algae (Leliaert et al. 2011) and ochrophytes (the topology presented in Figure 1). We present these data as heatmap analyses (Figure 4—figure supplement 2; Figure 8—figure supplement 2) with overviews within the main text (Figure 4, panel C; Figure 8, panel C). These analyses clearly support an origin for the ochrophyte green genes within the chlorophytes (rather than as a sister to all green genes), and for the haptophyte proteins with the pelagophyte/dictyochophyte clade. We additionally verify this latter position via a combined BLAST and tree sister-group analysis in Figure 8—figure supplement 3, and have revised the text surrounding this accordingly.

We additionally present an analysis of residues that are uniquely shared between ochrophytes and green algae, and between haptophytes and ochrophytes, in a new Figure 4, panel D, Figure 8, panel D, and Table S7). These analyses carry advantages over concatenated trees in that they neither require congruence in unrelated branches within the tree (as they trace the evolutionary relationships between two groups only, rather than considering global relationships between multiple groups), nor do they seek to ascribe a single origin point for a particular transfer (but rather tests whether multiple transfer events from different lineages have occurred). To avoid any possible artifacts within the analysis resulting from lateral gene transfer, paralogy or contamination, we have erected "best sets" consisting of 32 ochrophyte proteins of green origin, and 37 haptophyte proteins of ochrophyte origin that appear to have been subsequently entirely vertically inherited by all major groups of eukaryotic algae considered. We additionally clean each dataset of any residual contamination using a compound BLAST pipeline, where we profile the evolutionary affinities of each sequence within the alignment to a complete uniref + MMETSP database, and to every other sequence in the alignment. We present both these analyses, and the rationale for employing them, in a new section in the Materials and methods ("Identification of uniquely shared residues in multigene HPPG alignments"), and a new supplementary Table S6.

The analyses of uniquely shared residues clearly demonstrate that the majority of the green signal in ochrophytes is from an ancestral lineage within the chlorophytes (as opposed to resulting from multiple gene transfer events from individual green lineages), and that there has been a transfer of proteins from the ochrophytes into the haptophytes (as the majority of the uniquely shared residues detected are found in pelagophytes, dictyochophytes and diatoms, hence originated in a common ancestor of this lineage and were then transferred into haptophytes, rather than originating in haptophytes and being transferred later into pelagophytes and dictyochophytes).

Contamination: First, the fact that this was done should be more clearly presented (in Results) because the extent of contamination is quite serious, particularly in MMETSP.

We have revised the manuscript to state the manifold ways in which contamination has been removed, e.g. in the initial evolutionary analysis "To minimise misidentification due to any residual contamination in individual sequence libraries, only sequences for which the first three or more BLAST hits resolved within the same lineage were deemed to be unambiguously related to that lineage".

Second, removing 100% identical pairwise hits is not stringent enough, strictly speaking. No mismatches above 95% or 97% should be used when comparing distantly related groups (this would not work for closely related species) because errors and un-removed sequencing adapters often show up as differences. Also, the screen for contaminations should ideally have been done at the nucleotide level. The authors do not need to redo the analysis, however it is important that it is clearly stated in the text that further controls downstream (e.g., careful manual evaluation of each single tree) would have identified any remaining contamination and not bias the results.

In fact the initial decontamination analysis was performed using nucleotide sequences, using a custom threshold identity for each species pair based on the frequency distribution of similarity scores in a whole library BLAST of each species pair. The statement in the first draft was an error in comprehension by the primary author of the paper, for which we apologise.

The decontamination method used in this analysis has recently been published (in Marron et al., 2016), to which we refer the readers in the Materials and methods. We have also added a paraphrased explanation of the pipeline: "Cross-contamination was removed from MMETSP transcriptomes as previously described… A summary of the number of contigs discarded is provided in Table S1- sheet 2, section 1".

Cryptic endosymbioses: The foreign gene signals (green prasinophyte and haptophyte/pelagophyte) that were identified are likely to be genuine. These results are viewed as being most consistent with cryptic endosymbioses, which is controversial primarily because the assumptions rely on weakly supported precedents (extent of EGT in secondary endosymbiosis) or assumptions where essentially no evidence exists (e.g., strong HGT from a single source is unlikely). Therefore, please keep the discussion about cryptic endosymbioses as provisional.

We have revised the Discussion to state that the data supports a "possible" cryptic endosymbiosis in each instance. We note that repeated lateral gene transfer from a single source, while explaining the phylogenetic congruence of the gene transfers, would not necessarily explain the bias in transferred genes towards encoding proteins of plastid function, and have noted this in the Discussion (third paragraph). We have also by the same logic discussed the possibility of a chimeric endosymbiont (e.g. a red alga containing genes of green algal origin), which would be another equally plausible explanation to the data observed other than cryptic endosymbiosis.

*Figures and figure references: The figure references are unclear and not in a uniform format. Here are several examples:*

*"(Figure 2—figure supplement 1; Table S6)" vs. "(Figure 2, panel C; Figure 2—figure supplement 2, Figure 2—figure supplement 3- Table S5)"*

*"(Figures S6-S8)" Where are these figures?*

*Within the figure legend of Figure 6—figure supplement 2: "(Figure 5; Figure S25)". Where is Figure S25?*

Within Figure 2—figure supplement 2: why does it say "Figure S4" next to panel H?

We have globally revised all figure references in the manuscript for consistency.

*The figures are also unclear and the descriptions insufficient. Here are a few examples:*

Figure 2, panel A: what does the grey and white color codes indicate? The legend shows violet color that does not occur in the figure. Panel B: black arrows point in between categories.

We have revised both figures (now part of the revised Figure 5) and removed these issues.

Figure 2—figure supplement 2: the figure stretches over 4 pages and is very inconvenient to look at, the pathways are difficult to recognize and the referee cannot find explanations for the abbreviations used, the color code used is unclear (e.g.: What does the dark blue-green box used for the Kynurenine aminotransferase represent? What do blue or turquois letters as opposed to black letters signify?).

We have created a new version of this figure (now Figure 5—figure supplement 1) which is spread over just one page. We have globally used full enzyme names rather than abbreviations in this figure, and manually checked each component for correct colouring.

Figure 3: Panel B, and C, and Figure 3—figure supplements 2-5: The font used in the various figures is too small. Text within alignments and trees is illegible. This referee could not find a legend to the color code used to assign taxa within phylogenetic trees to specific lineages.

We have moved all large scale (single gene trees) to the supplementary materials, where they can be shown at 100% magnification rather than at a greatly condensed size. We have also revised all main text and supplementary figures so that the smallest font size used is Arial point 10 bold, with Arial point 8 bold being used sparingly and only where figure packing constraints necessitate its inclusion. We have provided a phylogenetic colour code to all supplementary figure trees with the first tree (Figure 2—figure supplement 2).

*Supplements: There is an excessive use of supplemental tables in this manuscript. Many of these tables are cryptic and the interesting information is buried in them instead of being explained in the text. Here is one example:*

"We additionally noted that within our dataset there were several HPPGs which have been previously annotated to function in organelles other than the plastid (for example, twenty HPPGs of annotated mitochondrial function, and four each of annotated endosomal and peroxisomal functions; Table S5). […] Other proteins, which lack any obvious homologues within plastid proteome data, might contribute entirely novel biological functions to the ochrophyte plastid proteome (Tables S4, S5)." What type of proteins are we looking at here and which novel biological functions could they contribute?

We have revised the main text so that all explicit points made within the text are supported directly by figures within the text (e.g. chi-squared P values) or refer to data presented within a figure. Where possible we have added data labels to each figure to allow rapid visual assessment of absolute values. We have removed the above example from the text and replaced it with a statement directly viewable from a figure: "106 of our inferred ancestral HPPGs include a *P. tricornutum* protein with prior experimental plastid localization, or unambiguous plastid function (Figure 1, panel D), but the remainder do not".

We have revised each of the supplementary tables, and the associated references in the main text, to make them more navigable to non-specialists (see discussion below).

All supporting datasets have been made publicly accessible through a third-party server (University of Cambridge dSpace) as discussed in the “Data Deposition” section of the Materials and methods.

Main text: Not all claims made in the text are sufficiently supported by the data. Here are several examples:

The referee would like to see wild type controls for the fluorescence microscopic images.

We have provided these controls in a new Figure 2—figure supplement 7, and Figure 7—figure supplement 1.

"In a majority (304/551) of trees for which plastid-targeted haptophyte homologues could be identified, haptophyte orthologues resolved within the ochrophytes (Figure 7, panel A). All other groups considered, including the alveolates, which have been proposed to possess a plastid of ochrophyte origin, more frequently branched external to than within the ochrophytes (Figure 7, panel A)." The text suggests that Figure 7, panel A resolves the phylogenetic positions of proteins within the ochrophytes which is not the case.

We have revised this statement to "In a majority (243/437) of trees in which they could be assigned a clear origin, plastid-targeted proteins from haptophytes resolved at a position within the ochrophyte clade (Materials and methods)" to make it clear that we are not grouping sequences in panel A by their specific evolutionary position within the ochrophytes, just whether they are positioned at any point within the ochrophyte clade. We also point the reader here towards the Materials and methods should they wish for more detailed definitions. We feel that this, in combination with the following statement that "we tabulated the individual ochrophyte sub-categories identified in the first sister-group to haptophyte sequences, of which the greatest number (Bown, 1998) resolved specifically with pelagophyte and dictyochophyte sequences (Figure 8, panel C)" [i.e. that panel C provides specific information about where in the ochrophytes these sequences are found] should be sufficient to imply that panel A only provides a very superficial, global, "inside or outside" definition.

"…However, these resolved as a separate clade to the C-type haptophyte and cryptomonad plastid rpl36 sequences, which formed a monophyletic clade with consistent support (Figure 8, panel D; consensus support > 0.5 in all Bayesian analyses), and were unified by three uniquely shared residues that were absent from all ochrophyte sequences (Figure 8, panel D)." The referee did not see a monophyletic clade of haptophyte and cryptomonad rpl36 sequences in the tree. The tree does not resolve the position of haptophytes/cryptomonads relative to dinoflagellates. Furthermore, a Bayesian probability >0.5 cannot be considered as consistent support. Finally, one of the three "uniquely shared" residues in the alignment that unifies the haptophytes is not shared by Pavlova.

Per the above, we have removed all analyses with *rpl36* from the Results section.

References:

1) Dittami, S.M., Michel, G., Collén, J., Boyen, C. & Tonon, T. Chlorophyll-binding proteins revisited--a multigenic family of light-harvesting and stress proteins from a brown algal perspective. BMC Evol Biol 10, 365 (2010).

2) Frommolt, R. et al. Ancient Recruitment by Chromists of Green Algal Genes Encoding Enzymes for Carotenoid Biosynthesis. Mol Biol Evol 25, 2653-2667 (2008).

3) Coesel, S., Obornik, M., Varela, J., Falciatore, A. & Bowler, C. Evolutionary origins and functions of the carotenoid biosynthetic pathway in marine diatoms. PLoS One 3, 2896 (2008).

4) Chan, C.X. et al. Red and green algal monophyly and extensive gene sharing found in a rich repertoire of red algal genes. Curr Biol 21, 328-333 (2011).

5) Dorrell, R.G. & Howe, C.J. What makes a chloroplast? Reconstructing the establishment of photosynthetic symbioses. J Cell Sci125, 1865-1875 (2012).

6) Morse, D., Salois, P., Markovic, P. & Hastings, J.W. A nuclear-encoded form II RuBisCO in dinoflagellates. Science 268, 1622-1624 (1995).

7) Matari, N.H. & Blair, J.E. A multilocus timescale for oomycete evolution estimated under three distinct molecular clock models. BMC Evol Biol 14, 101 (2014).

8) Brown, J.W. & Sorhannus, U. A molecular genetic timescale for the diversification of autotrophic stramenopiles (Ochrophyta): substantive underestimation of putative fossil ages. PLoS One 5, 12759 (2010).